# An age scale for new climate records from Sherman Island, West Antarctica

Isobel Rowell[1], Carlos Martin[2], Robert Mulvaney[2], Helena Pryer[1], Dieter Tetzner[2], Emily Doyle[1], Hara Madhav Talasila[3], Jilu Li[3], and Eric Wolff[1]

[1]Department of Earth Sciences, University of Cambridge, Downing Street, Cambridge, CB2 3EQ
[2]British Antarctic Survey, High Cross, Madingley Road, Cambridge, CB3 0ET
[3]Center for Remote Sensing and Integrated Systems, University of Kansas, 2335 Irving Hill Road, Lawrence, KS 66045

**Correspondence:** Isobel Rowell (ifr21@cam.ac.uk)

**Abstract.** Few ice cores from the Amundsen and Bellingshausen Sea sectors of the West Antarctic Ice Sheet (WAIS) extend back in time further than a few hundred years. The WAIS is believed to be susceptible to collapse as a result of anthropogenic climate change and may have at least partially collapsed during the Last Interglacial (LIG). Understanding the stability of the WAIS during warm periods such as the LIG and Holocene is important. As part of the WACSWAIN project, the British
Antarctic Survey's (BAS) Rapid Access Isotope Drill (RAID) was deployed in 2020 on Sherman Island in the Abbott Ice Shelf, West Antarctica. We drilled a 323 m deep borehole, with discrete samples of ice chippings collected covering the entire depth range of the drilled ice. The samples were analysed for stable water isotope composition and major ion content at BAS from 2020-2022. Using annual layer counting of chemical records, volcanic horizon identification and ice modelling, an age scale for the record of 1724 discrete samples is presented. The Sherman Island ice record extends back to greater than 1240 years before
present, providing the oldest, continuous, ice-derived palaeoclimate records in the coastal Amundsen and Bellingshausen Sea sectors to date. We demonstrate the potential for recovery of a complete Holocene climate record from Sherman Island in the future, and confidence in the ability of RAID samples to contain sufficiently resolved records for meaningful climatic interpretation.

## 1  Introduction

The West Antarctic Ice Sheet (WAIS) is believed to be vulnerable to collapse due to anthropogenic warming, with the potential to contribute several metres to global sea level (Mercer, 1978; Oppenheimer, 1998; Edwards et al., 2019; Bamber et al., 2019; Lowry et al., 2021). Much of the WAIS is grounded in marine basins lying up to 2000 m below sea level, making it highly susceptible to mass loss as a result of marine ice sheet instability, induced by ocean warming (Joughin et al., 2014; Shepherd et al., 2004). The recent thinning of WAIS and Southern Antarctic Peninsula ice shelves has resulted in accelerated flow of
their respective glaciers into the Bellingshausen and Amundsen Sea embayment and their grounding lines have subsequently retreated over the last few decades (Konrad et al., 2018; Paolo et al., 2015; Wouters et al., 2015). Current loss of ice volume from the WAIS is dominated by loss from the Pine Island and Thwaites Glaciers (Pattyn and Morlighem, 2020). Significant retreat of WAIS glaciers is underway and is identified as one of the major "tipping points" in the climate system, which could

bring about irreversible WAIS collapse (Lenton et al., 2008). WAIS loss has implications for other parts of the climate system, such as Antarctic sea ice coverage, albedo and freshening of the Southern Ocean, highlighting the importance of research in this area (Bronselaer et al., 2018; Wunderling et al., 2020).

The WAIS is believed to have at least partially collapsed during past warm periods, including the Last Interglacial (LIG) (Dutton et al., 2015). Investigating the behaviour of presently sensitive WAIS regions during these times provides insights into ice sheet stability which can be used to understand and predict current and future change. The WACSWAIN project (WArm Climate Stability of the West Antarctic ice sheet in the last INterglacial) aims to use ice core records to investigate the WAIS during the LIG, to supplement existing modelling studies (e.g. DeConto and Pollard (2016)). An ice core from Skytrain Ice Rise (Figure 1) was successfully drilled to the bed at 651 m in 2018-19 (Mulvaney et al., 2021), the results from which are now being published (Mulvaney et al., 2023; Hoffmann et al., 2022).

The second candidate site, Sherman Island, is located in the Abbott Ice Shelf between continental Antarctica and Thurston Island, close to the Thwaites and Pine Island glaciers (Figure 1 and Table 1). If present, LIG ice from Sherman Island would provide a second constraint of LIG WAIS stability through examination of stable water isotope records which could indicate the temperature and elevation history of the site and the palaeoclimatic variability of the Amundsen and Bellingshausen Sea sectors of the WAIS. Furthermore, a LIG record from this site, in addition to those from Skytrain Ice Rise and the upcoming Hercules Dome ice core further south (Fudge et al., 2022; Dütsch et al., 2023) and LIG data from the more westerly Mount Moulton (Korotkikh et al., 2011; Steig et al., 2015), would result in a more complete picture of the WAIS from this time. Sherman Island lies in what was predicted to be a region effectively rain-shadowed by the mountains on Thurston Island to the north, lowering the estimated accumulation rate at the site in comparison to nearby coastal WAIS ice rises. Ice sheet thickness data from two Operation IceBridge flyovers (IRMCR2, 2009) indicated ice of approximately 420 m depth on the island and estimates of accumulation rate and geothermal heat flux indicated the possibility of ice from the LIG towards the bed (Mulvaney et al., 2021). Due to the low-lying position of Sherman Island, it is possible that the dome could have been overridden during the Last Glacial Maximum, removing older ice.

The British Antarctic Survey's Rapid Access Isotope Drill (RAID) was used instead of carrying out a full-scale ice core drilling campaign, to mitigate the risks of a higher-risk site such as Sherman Island (Mulvaney et al., 2021). The RAID is a novel drilling technique which uses a single barrel to drill a dry borehole, obtaining stratigraphically ordered samples of ice chippings rather than a solid ice core. The samples can be discretely sampled and analysed to obtain a comparatively lower resolution record of measurements including stable water isotope composition (Rix et al., 2019). Drilling with the RAID progresses approximately three times more quickly than traditional drilling techniques and places a significantly lower logistics demand on projects, with the possibility of field setup, drilling and de-camping to completion within a few weeks. It has previously been demonstrated that the drilling and sampling techniques necessary for RAID ice do not result in significant mixing or attenuation of stable water isotopic or chemical signals in the ice (Nguyen et al., 2021; Rowell et al., 2022). Measurements of chemical concentration, for which the RAID was not initially designed, are climatologically meaningful and an identified chemical contamination problem does not impact climatic interpretation, particularly on longer time-scales (Rowell et al., 2022).

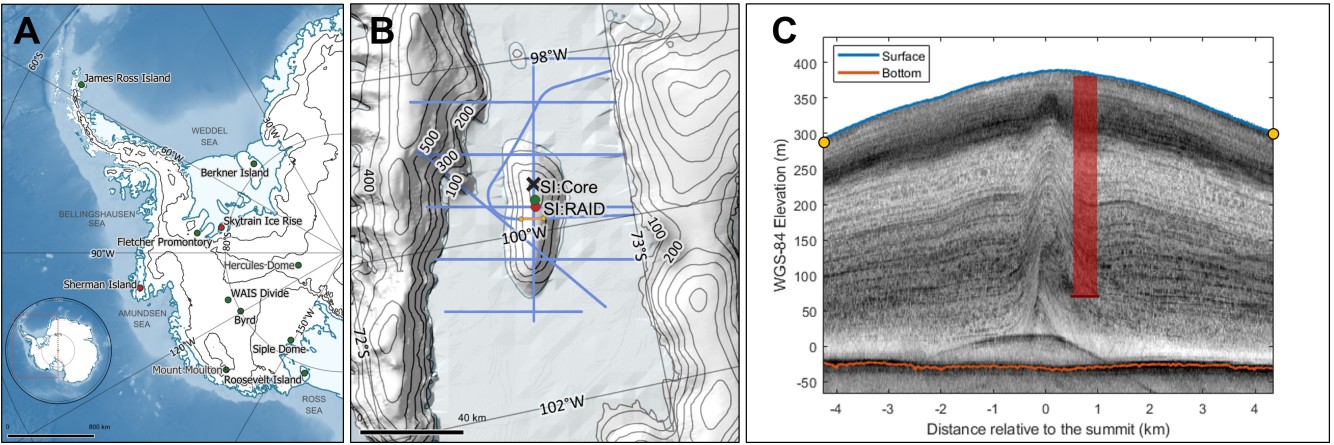

**Figure 1.** Panel A: map of West Antarctica indicating the location of the WACSWAIN drilling sites in red and other WAIS deep ice core sites in green. Panel B: close up map of Sherman Island, showing IceBridge flyover lines in blue, the line from which the echogram (Panel C) is taken in yellow, the SI:RAID site in red, the SI:Core site in green and the black X indicating the deepest location on the island (discussed in the text). Maps generated using QGIS with the Quantarctica mapping environment (Matsuoka et al., 2021). Contour lines show the elevation at 1000 m (panel A) and 100 m (panel B) intervals from CryoSat-2. Panel C: Radar echogram from the IceBridge line in yellow in Panel B, with the red column showing the approximate relative location of the SI:RAID drilling site to the ice divide.

**Table 1.** Information about the Sherman Island RAID and ice core sites and drilling campaign. SI:Core data from (Tetzner et al., 2022a) and IceBridge data.

| Core type | Latitude (°) | Longitude (°) | Elevation (m) | Ice thickness (m) | Depth drilled (m) | No. samples |
|-----------|-------------|---------------|---------------|-------------------|-------------------|-------------|
| SI:RAID | -72.67 | -99.71 | 440 | 428 | 323.23 | 1724 |
| SI:Core | -72.67 | -99.63 | 474 | ~434 to 437 | 21.3 | 425 |

Drilling on Sherman Island in early 2020 reached a depth of 323 m, out of a total ice sheet depth of approximately 428 m, at which point the drill became stuck in the ice and the drilling campaign ended (Mulvaney et al., 2021). Not reaching the bedrock is expected to significantly reduce the total possible length of the climate records obtained, due to the exponential increase of age with depth. For this reason and to maximise the potential use of the Sherman Island ice samples, the project aims turned to investigating the natural climate variability of the last few centuries of this vulnerable region of the WAIS, to set current and future warming and ice sheet behaviour into recent context. Crucial to this task is assigning a reliable age scale to the ice, and this paper addresses that for the new RAID core from Sherman Island. Using the ice we do have, we also assess the oldest ice which may be available from this site from a deeper core towards the bed.

## 2 Methods

### 2.1 Drilling and Measurements

The BAS RAID (Rix et al., 2019) drilled on Sherman Island to a depth of 323 m in five drilling days, collecting 1724 discrete samples of ice chippings at a resolution of 6 to 29 cm (average 19 cm) as described in Rowell et al. (2022) and Mulvaney et al. (2021). Stable water isotope composition ($\delta^{18}O$ and $\delta^2H$) and chemical ion concentrations ($Ca^{2+}$, $K^+$, $Mg^{2+}$, $Na^+$, $Cl^-$, $SO_4^{2-}$, $MSA^-$, and $NO_3^-$) were measured on the samples at BAS from 2020 to 2021. The $Ca^{2+}$ and $K^+$ concentrations in the top-most sample of each drop of the drill into the ice was found to be artificially high. After discarding the contaminated data, the remaining dataset appears robust and suitable for investigating the annual layers and trends in the concentrations of chemical ions and stable water isotope composition (Rowell et al., 2022).

### 2.2 Age scale development

The Sherman Island RAID (SI:RAID) age scale was produced using three methods. Annual layer counting of stable water isotopic and chemical species was carried out on the basis that the stable water isotopic composition and concentration of certain ions vary seasonally, enabling the visualisation and counting of peaks and troughs corresponding to one annual cycle, or layer, in the ice (e.g., Sigg and Neftel (1988); Sigl et al. (2016); Winstrup et al. (2019)). The layers can simply be counted from one year to the next, giving fixed depths for the summer and/or winter of the shallower, more recent years in the ice sheet. Identification of large, well-dated volcanic events in the sulfate ($SO_4^{2-}$) record, supplemented with sulfur (S) isotope analysis to differentiate between background and volcanic samples, was conducted to date ice beyond the depth where annual layer counting was possible (Patris et al., 2000). Two ice thinning models were used to give an initial estimate of the age scale, assess confidence in the first two dating methods and provide an age estimation for the deepest ice. The three steps are described in further detail below.

#### 2.2.1 Annual layer counting

Annual layer counting (Figures 2 and 3) was performed on stable water isotope ratio and chemical data plotted on a depth scale. The MATLAB programme "Matchmaker" was used for plotting multiple records together: from the SI:RAID data and a nearby 20 m long Sherman Island ice core (hereinafter referred to as SI:Core). Matchmaker was used to identify corresponding peaks and troughs, place markers and adjust the age scale accordingly (Rasmussen et al., 2013; Tetzner et al., 2022b, 2021). The species used consistently throughout annual layer counting were: deuterium ($\delta^2H$), methanesulfonate ($MSA^-$), sulfate ($SO_4^{2-}$) and sodium ($Na^+$). These species typically have highest values during the austral summer (December to February) because $\delta^2H$ responds primarily to temperature and $SO_4^{2-}$ and $MSA^-$ are related to marine bio-productivity, both of which peak in the summer (Turner et al., 1995). The relative contribution of $Na^+$ seasonally to the ice appears to be inconsistent at Sherman Island, with some years seeing summer peaks corresponding to clear summer signals in the $\delta^2H$ record and other years showing winter peaks. This variability could reflect the local geography of Sherman Island, which is close to open water

sea salt contributions in the summer and is then closely surrounded by salt-producing sea ice in the winter, with potential contributions from the nearby Pine Island and Amundsen Sea polynyas (Tetzner et al., 2019). The contribution of $Na^+$ to the

100 ice core site could therefore be dependent on wind direction and circulation patterns rather than seasonality. Counting was therefore supplemented with the $SO_4^{2-}/Na^+$ ratio in places where the seasonality of $Na^+$ was unclear.

Annual age markers from the SI:Core were used to add summer-to-summer reference points in the top 20 m of the RAID data to aid the layer counting. The SI:RAID data show similar absolute concentrations to the SI:Core, as well as closely matching annual variability throughout much of the 20 m, but with less well-defined peaks because of the lower depth resolution (Figure

2). Using the spacing of the annual layers in the top 20 m, with the SI:Core as a guide, counting was continued to 70 m using the same technique as for the top 20 m but without the assistance of the SI:Core data. It is possible that annual layer counting could have been continued below 70 m. However, the regular variations, which arguably are still of seasonal origin, are difficult to distinguish from variability caused by other factors. In particular, seasonal variability in $\delta^2 H$ becomes limited below this depth. The regular peaks and troughs that are indicative of seasonal variations are formed from typically less than four data

points below 70 m meaning that annual layer counting below this point cannot be considered robust. Annual layer counting data (summer peaks) are available in Supplement S1.

### 2.2.2 Modelling ice thinning and *a priori* estimate of depth/age

Using the age scale for the top 70 m from annual layer counting, it was possible to estimate the age of deeper samples. This estimate was done in combination with an ice thinning model. Details about the model can be found in Martín et al. (2015),

but we summarize here the main equations for convenience. The model neglects horizontal advection and calculates age $A$ at a given depth $d$ and time $t$ as a function of the vertical velocity $w$ as shown in Equation 1:

$$\frac{\partial A}{\partial t}(d,t) - w(d,t)\frac{\partial A}{\partial d}(d,t) = 1 \qquad 0 \leq d \leq H, \quad -t_0 \leq t \leq 0$$
$$A(d,t0) = A_0,$$
$$A(0,t) = 0, \tag{1}$$

where $H$ is the ice thickness, and $A_0$ is the initial depth/age, a time $t_0$ before the present $t = 0$.

We further assume that in the vertical velocity we can separate the time- and depth-dependency and that there is no basal

melt or variation of ice thickness with time. The assumption of no basal melt is based on the estimated basal temperature at Sherman Island of -6 °C, from borehole temperature measurements (Mulvaney et al., 2021). The vertical velocity can be then written as shown in Equation 2:

$$w(d,t) = -a(t)\frac{\rho_i}{\rho(d)}\eta(d), \tag{2}$$

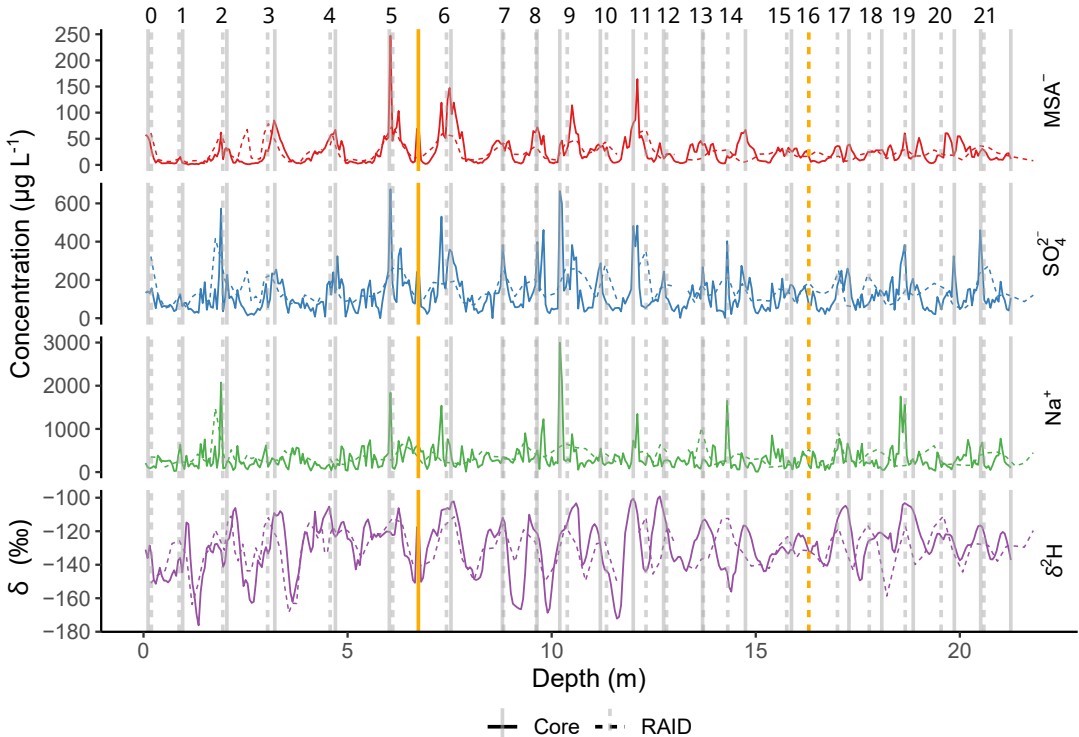

**Figure 2.** Annual layer counting of the top 20 m of the Sherman Island MSA⁻, SO₄²⁻, Na⁺ (in μg$L^{-1}$) and $\delta\,^2$H (‰) data. Counts are performed using both SI:RAID (dashed lines) and SI:Core (solid lines) data. Grey vertical age markers are placed on the assigned summer peak. Yellow lines represent "uncertain" years in both core and RAID and are used in assessing the age scale uncertainty, described in the text. The numbers along the top represent the number of years counted from the surface.

where $\rho$ is the density, that is assumed uniform in time, $\rho_i$ is the density of ice, $a$ is the time-dependent annual accumulation, and $\eta(d)$ is a function of depth, often referred as the shape function, that varies between 0 at the bed and 1 at the surface. For the shape function we use two extreme approximations. For one extreme, we use Lliboutry (1979), Equation 3:

$$\eta(d) = 1 - \frac{p+2}{p+1}\left(\frac{d}{H}\right) + \frac{1}{p+1}\left(\frac{d}{H}\right)^{p+2}, \tag{3}$$

where $p$ is a parameter. The Lliboutry approximation reproduces well the ice flow dominated by shear (Martín and Gudmundsson, 2012) and we use it to simulate ice thinning at the flanks of the ice flow divide. "Flank" describes the flow located more than a few thicknesses away from the ice flow "divide", which is often located near the ridge perpendicular to ice flow (Figure 1, Panel C). On the other extreme, to represent ice flow divide conditions, we use the numerical output at the divide from the full field model of Martín and Gudmundsson (2012).

For our *a priori* depth/age model we assume that the accumulation rate is proportional to that of WAIS Divide (Sigl et al., 2016) using the present values of annual accumulation rate at Sherman Island estimated from annual layer counting. Our

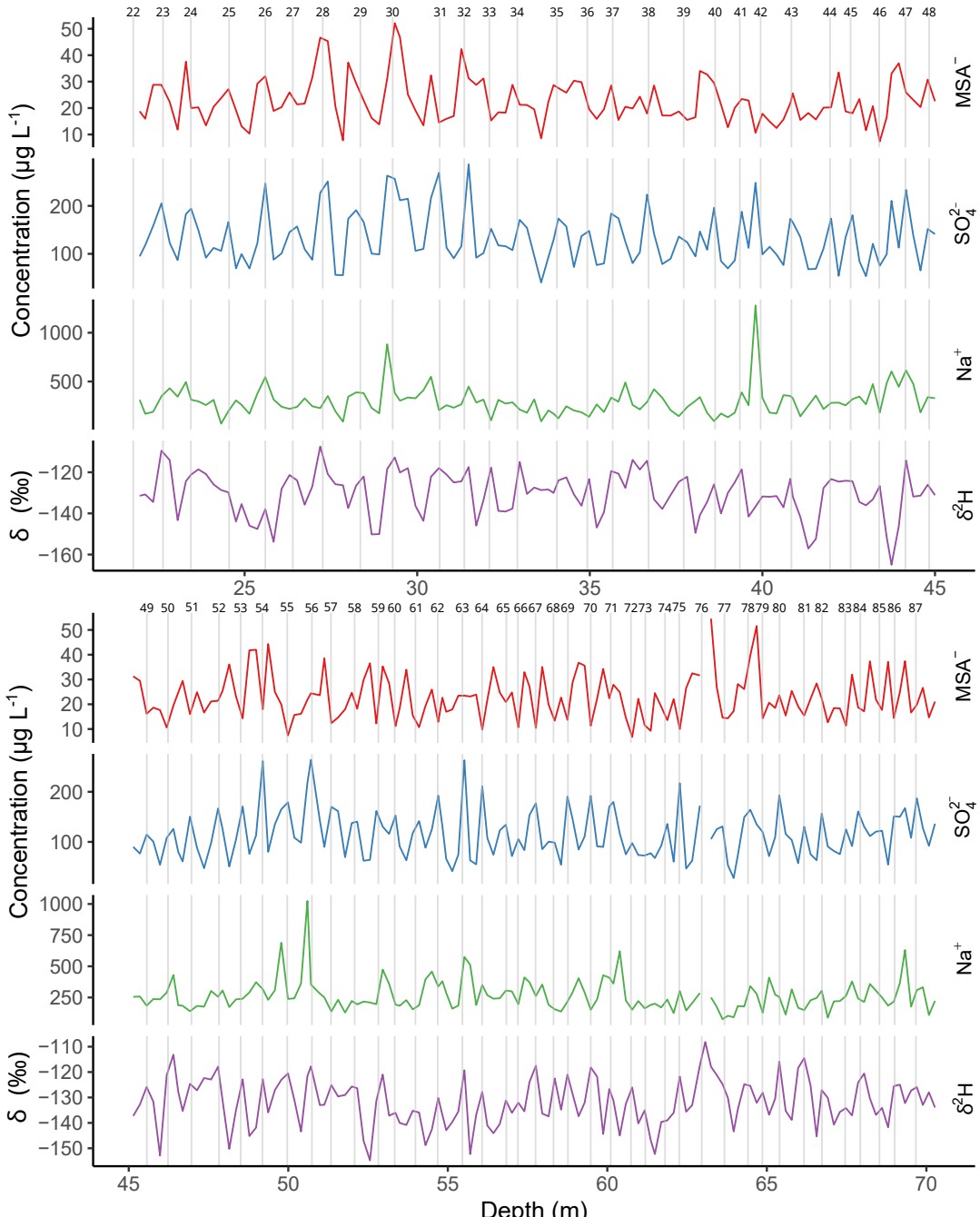

**Figure 3.** Annual layer counting from 20 to 70 m of the SI:RAID MSA$^-$, SO$_4^{2-}$, Na$^+$ (in µg$L^{-1}$) and $\delta\,^2$H (‰) data. Grey age markers are placed on the assigned summer peak. Y-axis ranges of panels adjusted for the range of concentrations. The numbers along the top represent the number of years counted from the surface.

hypothesis is that the flow conditions can be bounded between divide flow and flank flow with $p = 3$. The results for the depth/age estimation from both extremes are shown in Figure 5. The annual layer counting (yellow markers in Panel A of Figure 5) aligns more closely with the depth/age model for flank-flow conditions than divide-flow, by the bottom of the annual layer counts. We therefore use the flank-flow depth/age model to guide us in dating the deeper samples, as described below.

### 2.2.3 Volcanic horizon identification

The records of chemical species were closely inspected for signatures of volcanic events as explained below. Based on the *a priori* depth/age model, an average annual layer thickness of 87 cm in the top 70 m, and with a 19 cm average sample resolution, samples should continue to be annual, or greater, resolution until at least 250 m depth. Assuming the imprint of volcanic events is recorded in the $SO_4^{2-}$ concentration at Sherman Island, the age resolution of the samples is therefore sufficient for resolving individual volcanic events.

Large volcanic eruptions emit sulfurous gases such as sulfur dioxide ($SO_2$) and hydrogen sulfide ($H_2S$) into the atmosphere, which are oxidised and precipitated onto the ice sheets in the form of $SO_4^{2-}$, leaving a peak in the $SO_4^{2-}$ concentration of the ice layer (Delmas et al., 1985). Matching volcanic peaks in the $SO_4^{2-}$ record of ice cores with known volcanic events provides an age constraint at certain depth intervals (e.g., Udisti et al. (2004); Parrenin et al. (2012); Severi et al. (2012)). The Sherman Island drill site is only 440 m above sea level and located very close to open ocean. The $SO_4^{2-}$ record is therefore dominated by marine biogenic and sea salt sources, making volcanic peaks more difficult to identify. The total $SO_4^{2-}$ concentration was split into its sea salt (ss) and non sea salt (nss) components to aid the identification of volcanic peaks, according to Equations 4 to 8.

$$[\text{nssCa}^{2+}] = [\text{Ca}^{2+}] - [ss\text{Na}^+] \cdot \text{R}_\text{m} \tag{4}$$

$$[ss\text{Na}^+] = [\text{Na}^+] - [nss\text{Ca}^{2+}] \cdot \text{R}_\text{t} \tag{5}$$

$$[\text{nssSO}_4^{2-}] = [\text{SO}_4^{2-}] - \text{R}_\text{s} \cdot [ss\text{Na}^+] \tag{6}$$

with

$$\text{R}_\text{m} = (\text{Na}^+/\text{Ca}^{2+})_{ss}^{-1} \tag{7}$$

and

$$\text{R}_\text{t} = (\text{Na}^+/\text{Ca}^{2+})_{nss}, \tag{8}$$

where $R_m$ and $R_t$ are the $Ca^{2+}$:$Na^+$ sea salt and continental ion mass ratios respectively and $R_s$ is the sea salt ion mass ratio for $SO_4^{2-}$. Values of 1.78, 0.038 and 0.25 were used for $R_t$, $R_m$ and $R_s$, respectively (Röthlisberger et al., 2002; Bigler et al., 2006). As described in Rowell et al. (2022), to make best use of the $Ca^{2+}$ data while avoiding contaminated or missing data, the $Ca^{2+}$ data points for the first sample in each drop of the drill (which is generally contaminated) were replaced with the mean of the rest of the drop. This correction enables use of the $Ca^{2+}$ record to obtain $ssNa^+$ and subsequently $nssSO_4^{2-}$ apportionments.

The concentrations of $MSA^-$, $SO_4^{2-}$ and $nssSO_4^{2-}$ were plotted together (Figure 4) to identify potential volcanic peaks in the $SO_4^{2-}$ record. $MSA^-$ is used to corroborate the $nssSO_4^{2-}$ record, by providing a reference record of purely marine biogenic origin, to assist in the identification of $nssSO_4^{2-}$ peaks that are not related to fluctuations in marine inputs (Saigne and Legrand, 1987). Several eruptions known to leave a signal in the $SO_4^{2-}$ records of multiple ice cores from across Antarctica were targeted. Multiple viable peaks in the $SO_4^{2-}$ record, including some not related to these targeted events, were identified in approximately the expected depth range for these events as estimated from the initial age/depth model. Due to the relatively high background concentration of $SO_4^{2-}$ in the SI:RAID samples, S isotope analysis was used to identify $SO_4^{2-}$ peaks that have a volcanic source, which increases our confidence in identifying specific volcanic events in the $SO_4^{2-}$ data.

Sulfur (S) has four stable isotopes: $^{32}S$, $^{33}S$, $^{34}S$, $^{36}S$, which have natural relative abundances of 95.02%, 0.75%, 4.21% and 0.02%, respectively. Volcanic emissions have isotopically-light S compositions (i.e. relatively depleted in the $^{34}S$ isotope) compared to other dominant inputs of $SO_4^{2-}$ from marine biogenic emissions or sea salt (Rees et al., 1978; Nielsen et al., 1991; Patris et al., 2000; Crick et al., 2021). As such, the $\Delta^{34}S$ values (Equation 9) recorded in ice can be used to determine peaks in $SO_4^{2-}$ that have a volcanic source. The relative difference between the $\delta^{34}S$ and $\delta^{33}S$ values and the ratio expected from equilibrium fractionation is expressed using $\Delta^{33}S$ notation (Equation 10, in ‰). Values of $\Delta^{33}S$ that are outside analytical error of zero show mass-independent fractionation (Farquhar et al., 2001). Mass-independent fractionation of S isotopes occurs when sulfur dioxide gases are photo-oxidised to $SO_4^{2-}$ by short-wave UV radiation (Savarino et al., 2003). This process only occurs above the ozone layer meaning non-zero $\Delta^{33}S$ values indicate volcanic eruptions where the eruptive plume reached the stratosphere (Gautier et al., 2019; Burke et al., 2019).

$$\delta^x S(‰) = \left( \frac{(\delta^x S/\delta^{32}S)_{sample}}{(\delta^x S/\delta^{32}S)_{standard}} - 1 \right) \times 1000 \tag{9}$$

where x is either $^{34}S$ or $^{33}S$ and values are reported relative to the Vienna-Canyon Diablo Troilite standard.

$$\Delta^{33}S(‰) = \delta^{33}S - ((\delta^{34}S + 1)^{0.515} + 1) \times 1000 \tag{10}$$

A total of 75 samples from Sherman Island were analysed for their S isotope composition, using the method described in (Hoffmann et al., 2022), and 11 individual volcanic eruptions were identified (Figure 4). Peaks in the $SO_4^{2-}$ record defined as volcanic had $\delta^{34}S$ values ranging from 0.2 to 16.18 ‰ (mean = 11.18, $\sigma$ = 4.37, $n$ = 20), whereas background samples had $\delta^{34}S$ values ranging from 16.05 to 20.33 ‰ (mean = 18.23, $\sigma$ = 0.95, $n$ = 55). $\Delta^{33}S$ values were considered to indicate stratospheric eruptions if they were greater than 0.15 or less than -0.15 ‰. All S isotope data are provided in Supplement S2. Through

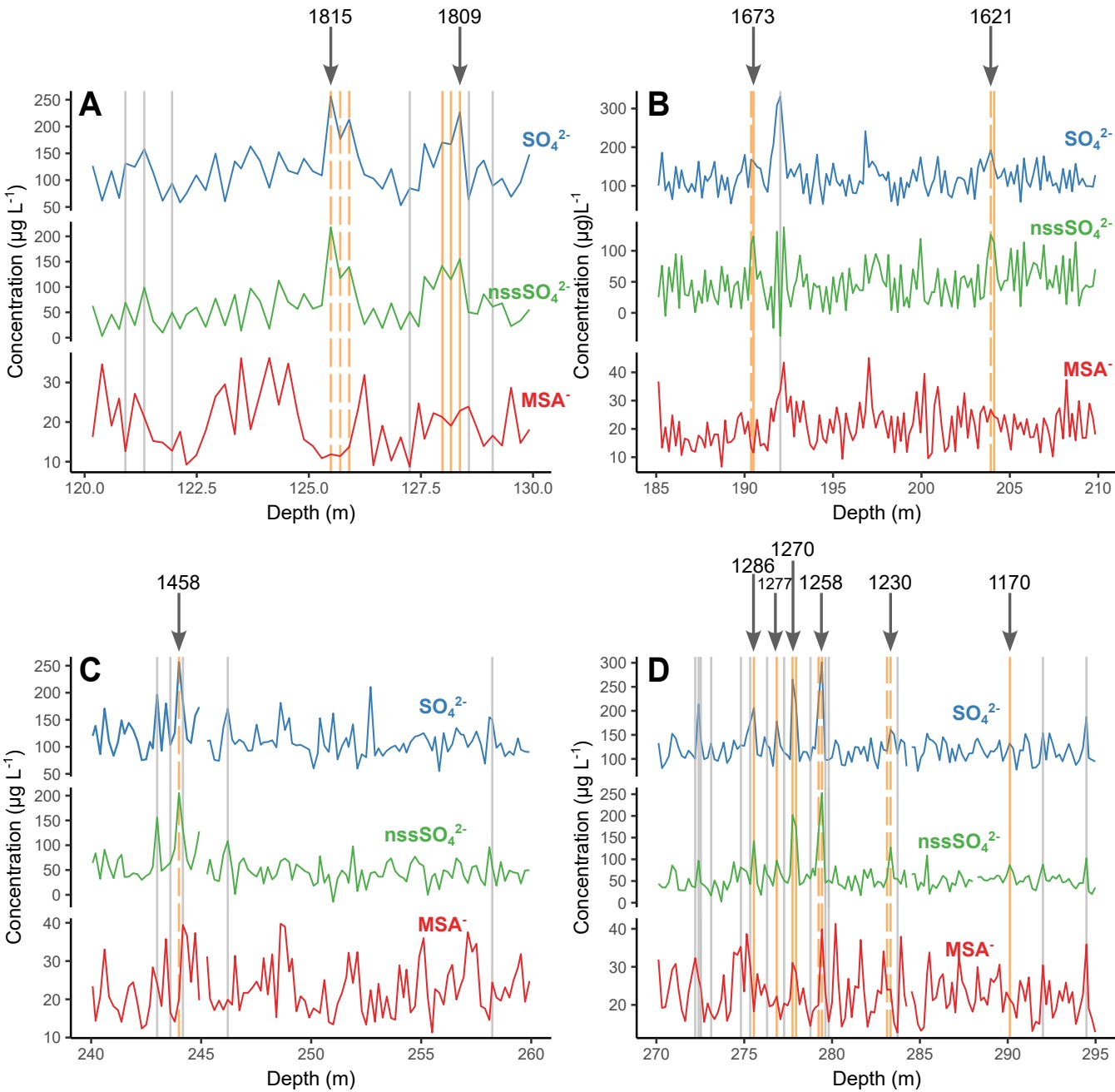

**Figure 4.** Identification of potential volcanic markers. The vertical lines show candidate peaks in the $SO_4^{2-}$ record that were analysed for S isotope compositions. Grey lines show the depths of samples that had background, non-volcanic S isotope compositions, yellow lines show the depth of samples that had low $\delta^{34}S$ values, indicative of volcanically-derived $SO_4^{2-}$ and dashed lines show volcanic peaks of stratospheric origin, defined by non-zero $\Delta^{33}S$ values. Panels A, B, C and D show the discrete depth ranges in which the identified volcanic markers appear. Arrows and numbers show the year of the identified event (CE).

a combination of comparison to the modelled age (Figure 5, Panel B), assessing the relative depth-age difference between eruptions and cross-checking against previously identified volcanic eruptions in Antarctic ice cores, the eruptions were dated. The $SO_4^{2-}$ peaks are believed to correspond to the eruption of Tambora in 1815, an eruption of unknown origin in 1809, an eruption in 1458 (possibly Kuwae), five eruptions of the thirteenth century (including Samalas) sequence (1286, 1277, 1270, 1258 and 1230), eruptions in 1621, 1673 and an eruption of unknown origin in 1170 (Sigl et al., 2014). Several eruptions

were defined by multiple samples and show the evolution of $\delta^{34}$S and $\Delta^{33}$S values during an eruptive event. Samples that had non-volcanic S isotope values were also measured between every volcanic event to allow for precise depth assignment and ensure that multiple eruptions were defined by separate peaks. 11 eruptions were used for the final the age/depth interpolation as described below.

### 2.2.4  Model optimisation and final depth/age interpolation

The deepest fixed age marker attributable to a known volcanic event is at 290 m depth (year 1170 CE). To date the remaining 33 m of ice samples and remaining ice below the drilled depth, we use a depth/age model that assimilates age markers and optimises past accumulation rate and ice flow parameters. The depth/age model is the result of an optimisation. We use a model that is identical to the one described in Section 2.2.2 but we find the values of accumulation rate variation $a(t)$ that provide a best fit between the model and the age markers. We assume that the accumulation rate history, $a(t)$, is a piece-wise linear

function and use multiple values for $p$, ranging from 1 to 4 to simulate the so-called "flank" flow. Under these assumptions we optimise the model using the Simplex method from Lagarias et al. (1998) as coded in the *fminsearch* function of *Matlab*. The results of the depth/age optimisation is shown, alongside the original model, in Figure 5 (Panel B).

The model optimisations using $1 \leq p \leq 4$ accurately captured the age of the volcanic tie points to within the uncertainty resulting from the discrete sample depth range. The selection of points for accumulation rate to change within the age/depth

optimisation model introduces a source of error: this is because although the tie points' locations necessitate a shift in accumulation rate, the exact timing of this change is unknown. This is the reason for assigning a relatively high uncertainty to the model age/depth model (10%, described below) and for using the mean of the flank age/depth relationships as the final interpolation beyond annual layer counting. The annual layer counts were used for interpolation of the age scale for the top 70 m. Below this depth, the mean of these optimisation simulations was used as the final age scale, with the range of values

used in defining the age scale uncertainty (described below). Due to the high resolution of the annual layer counts and model simulations, a simple linear interpolation between all points was assigned to each sample, with a top and bottom age associated with the top and bottom sample depths.

### 2.3  Radar data processing

The radar echogram (Figure 1, Panel C) was obtained using a Multichannel Coherent Radar Depth Sounder (MCoRDS,

Rodriguez-Morales et al. (2014)) from the IceBridge flight over Sherman Island on November 16th 2018, during the Bell Am Divide IS2 science mission in the 2018 Antarctica DC8 campaign. A larger aperture angle (115 degrees) was used in SAR (Synthetic Aperture Radar) processing by CReSIS toolbox to recover and enhance the echoes from the ice layers at the ice

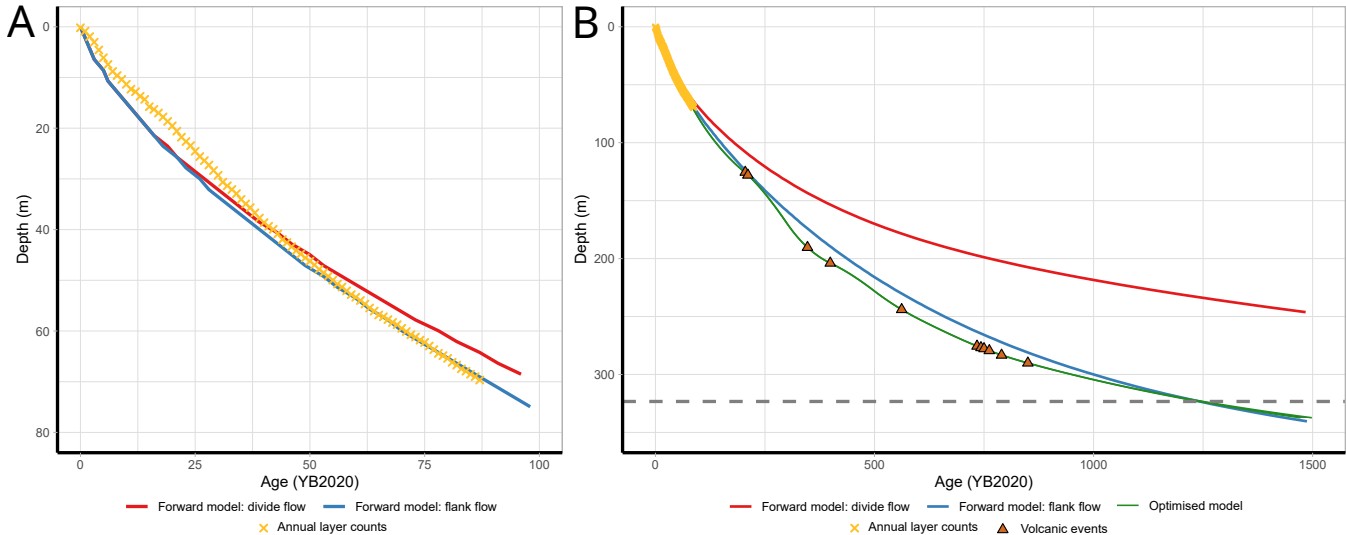

**Figure 5.** A: Depth/Age from the forward model of ice thinning, showing divide flow (red) and flank flow (blue), with annual layer counts (yellow crosses). B: original model flank and divide flows (red and blue lines, same as in panel A) and optimised model simulations (p = 1 to 4, green line) showing the tie points of annual layer counts (yellow crosses) and volcanic tie markers (brown triangles). The grey dotted line shows the bottom of the borehole and drilled ice.

divide (Paden et al., 2021). The echoes from these layers were missing or weak due to the slope effects ($\sim \pm 10$ degrees) in the echogram from the routine processing, which used an 18-degree full aperture angle. The data were decimated along track

after SAR processing by a factor of 26, using the average of the 51 range lines centered around the output range line, and thus resulting in 6.5 m between two neighboring range lines. The radar echogram was detrended using multiple polynomials to display the ice layers more clearly. The length of the echogram is $\sim$8.6 km, displayed relative to the summit (start-point 72.6132 °S and 99.8208 °W, end-point 72.6895 °S and 99.8576 °W) with an average aircraft altitude of 380 m above the ice surface. The ice thickness at the summit is $\sim$419 m assuming the ice dielectric constant is 3.15. The traced ice surface and

bed interface are delineated by blue and red lines, respectively. The strong interface close to the ice bed under the summit that follows the surface topography is the surface multiple reflections.

## 3 Results

### 3.1 Age scale and uncertainty

The depth/age profile, with uncertainty limits, of the SI:RAID data is shown in Figure 6. Depths and ages presented represent

the bottom of each sample, unless specified otherwise; the full age scale dataset with top and bottom depths and ages can be found in Supplement S2. The 323 m record of ice samples reaches an age of 1176 years before 1950, or the year 774 CE, $\pm$ 41 years. The age profile follows closely to idealised glacial flow, with small adjustments in accumulation rate necessary to

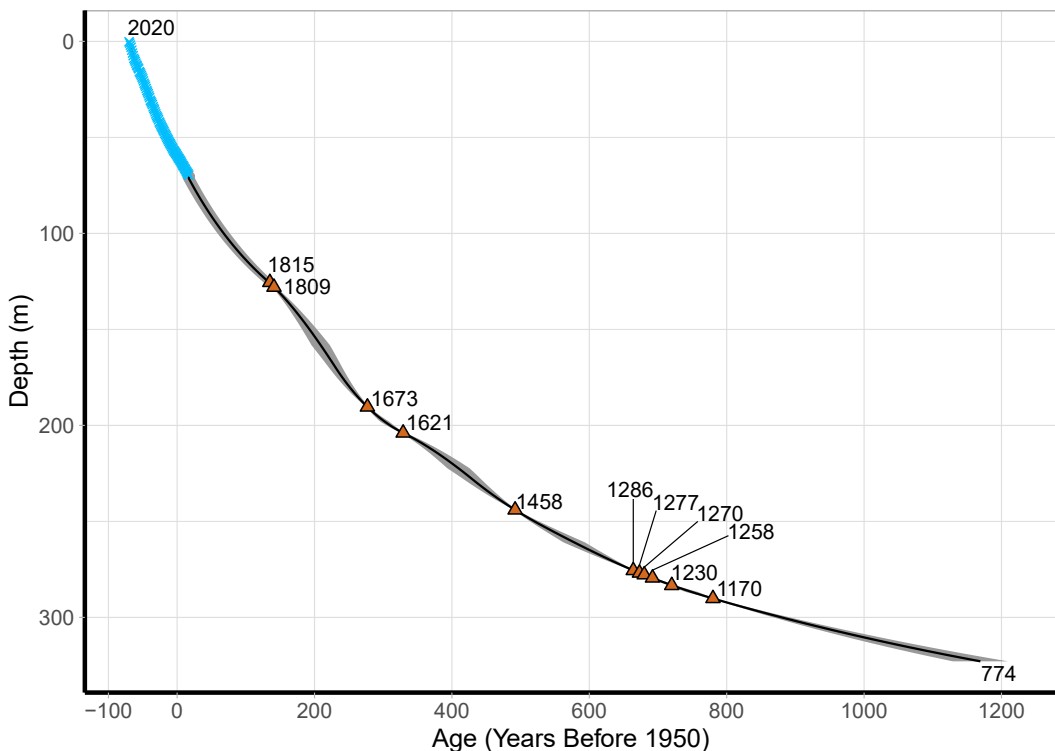

**Figure 6.** The SI:RAID age scale (black line), shown with the respective age markers (blue crosses are annual layer counts, orange filled triangles are volcanic events) used in its development and grey shading indicating uncertainty.

satisfactorily intercept every known age marker, as discussed below. The maximum absolute uncertainty in the whole record is 41 years at 323 m, an percentage uncertainty of 3.3% of the total age at that point. An uncertainty estimate was calculated for the annual counts, each volcanic tie point, the mid-points between each tie point, and finally for the model interpolation point, as described below.

The uncertainty associated with the annual layer counting was estimated using the SI:RAID and SI:Core annual counts together to assess the presence or absence of uncertain annual layers. Peaks present in some species in the RAID data which were not identified in the Core (or vice versa, shown by yellow bars in Figure 2), indicate uncertain years. There were two uncertain years out of 21 counted years for the top 21.8 m (when the SI:Core data ends), an uncertainty of 9.5 %. A 10 % uncertainty was therefore applied throughout the section that was annually counted in the RAID data (to 70 m).

For the volcanic tie points, we are confident about the identified volcanoes and their timing of eruption. Uncertainty at the tie point comes only from the depth interval of each eruption and its corresponding range of ages. Some of the identified volcanic events are recorded in the $SO_4^{2-}$ record over multiple samples. When this was the case, the sample with the highest $SO_4^{2-}$ concentration was chosen as the year of the event. Based on interpolation of the age scale to the point of each volcanic

**Table 2.** Estimates of age uncertainty for every tie point used in the SI:RAID age scale interpolation and the mid points between them (to nearest sample). Ages given in Year CE rounded to nearest year.

| Tie point type | Sample(s) | Bottom Depth (m) | Sample Bottom Age (CE) | Total uncertainty (years) |
|---|---|---|---|---|
| Top | 1 | 0 | 0 | 0 |
| Annual layer count bottom | 366 | 69.70 | 1933 | 9 |
| mid point | 532 | 99.87 | 1875 | 9 |
| 1815 Eruption | 668 | 125.49 | 1815 | 8 |
| 1809 Eruption | 682 | 128.17 | 1809 | 3 |
| mid point | 865 | 162.71 | 1741 | 14 |
| 1673 Eruption | 1014 | 190.37 | 1673 | 2 |
| mid point | 1051 | 197.22 | 1648 | 7 |
| 1621 Eruption | 1087 | 203.92 | 1621 | 3 |
| mid point | 1207 | 226.52 | 1540 | 17 |
| 1458 Eruption | 1301 | 243.99 | 1458 | 2 |
| mid point | 1393 | 261.03 | 1372 | 16 |
| 1286 Eruption | 1471 | 275.55 | 1286 | 2 |
| mid point | 1473 | 275.96 | 1281 | 4 |
| 1277 Eruption | 1478 | 276.85 | 1277 | 2 |
| mid point | 1480 | 277.27 | 1273 | 4 |
| 1270 Eruption | 1483 | 277.75 | 1270 | 3 |
| mid point | 1487 | 278.56 | 1264 | 5 |
| 1258 Volcano | 1492 | 279.42 | 1258 | 3 |
| mid point | 1503 | 281.50 | 1244 | 6 |
| 1230 Eruption | 1513 | 283.33 | 1230 | 4 |
| mid point | 1533 | 286.83 | 1201 | 9 |
| 1170 Eruption | 1551 | 290.11 | 1170 | 2 |
| Model | 1552 to 1726 | 290.31 to 323.3 | 774 to 1170 | 2 to 41 |

event, the age interval of that sample (and surrounding samples) was calculated. This gives an uncertainty for each volcanic tie point.

The optimised model age/depth fits through the volcanic tie points, as it was designed to do, within the uncertainties described above. An estimate of uncertainty for the remaining age scale is derived from the relative difference in age between the models
run with different $p$ parameters at 90 % depth, approximately 10 %. A 10 % uncertainty from the previous tie point is therefore applied to the mid-points between tie-points, added to the average uncertainty at the two adjacent ties (Table 2).

The age scale uncertainty was interpolated to every sample using a linear interpolation of the error between the points shown in Table 2, up to the last tie point (volcanic at 1170 CE). For all samples below this depth, the model uncertainty (10%) was applied as a percentage relative to the age difference from the final tie point (1172 CE), added to the uncertainty at that tie
point. All uncertainty values are shown in Table 2. The age scale (limited to the sample bottom depths and ages), is provided in Supplement S3.

## 3.2 Prediction of age near the bedrock

Predicting the age of the 105 m of ice below the bottom sample depth, where the drill was lost, is more uncertain. The optimisation model was used to estimate the age of the ice towards the bed at the RAID drilling site; however, in the deepest ∼5%, model outputs are highly dependent on input parameters and thus unrealistic. This is because in our depth/age model we assume no basal melting and the solution to Equation 1 is that the age of ice at the bottom tends to infinity. Numerically this translates into a solution at the bottom that depends on the numerical details used, mainly the unknown initial conditions. Instead we use, as a conservative estimate, the age at 90% depth (∼385 m). For the existing SI:RAID site, assuming accumulation rate varies approximately within modern values and relative to WAIS Divide prior to 774 CE, it is likely the age at 90% depth is between ∼ 3,100 and 3,400 years before 1950, extending to approximately 6 to 7.1 ka at 95% depth (∼406 m). Our expectation is that the depth of older features, such as the early Holocene or Last Glacial Maximum (LGM), if present, are likely on the order of a maximum of a few metres above the bed.

A final attempt to locate older ice on the island used the accumulation rate histories as calculated for simulations of the SI:RAID site age/depth. The piecewise values of $a(t)$ (Equation 2) are used as inputs for the age/depth optimisation, with age/depth tie points removed and $p = -1$ (in practice -0.99), to assume divide flow conditions as described in Section 2.2.2. To test the impact of the maximum age given to the model, two values (150 kyrs and 25 kyrs) were used. The results are shown in Figure 7.

## 3.3 Seasonality of chemical species

The species used for annual layer counting were interpolated to monthly resolution for their entire records. Monthly mean anomaly concentrations, relative to annual means, were then calculated to investigate the seasonality of species with depth, using methods similar to Hoffmann et al. (2022) (Figure 8). All species show strong seasonality in the top 20 years, which were annually counted alongside SI:Core. $SO_4^{2-}$ and $\delta^{18}O$ demonstrate this continued seasonality consistently to 70 m (annually counted). $MSA^-$ also demonstrates a seasonal pattern throughout this period, but with a changed seasonality from a summer peak to a slight winter peak, in line with $Na^+$. This migration of the $MSA^-$ peak has been well documented in ice cores, and in the SI:RAID samples begins at approximately 30 m, with inconsistent seasonality in peaks for 15 m followed by consistent winter peaks (corresponding with troughs in $\delta^2H$) by 46 m depth (Osman et al., 2017; Pasteur and Mulvaney, 2000; Curran et al., 2002).

## 4 Discussion

### 4.1 Dating methods

The Sherman Island RAID samples present a relatively low resolution ice core record due to the sampling restraints to keep cargo minimal. The samples average 19 cm depth resolution, in age increments ranging from 0.04 to 2.38 years with an average of 0.7 years. This presents a challenge for dating compared to other ice cores; traditional ice cores can be sampled and measured

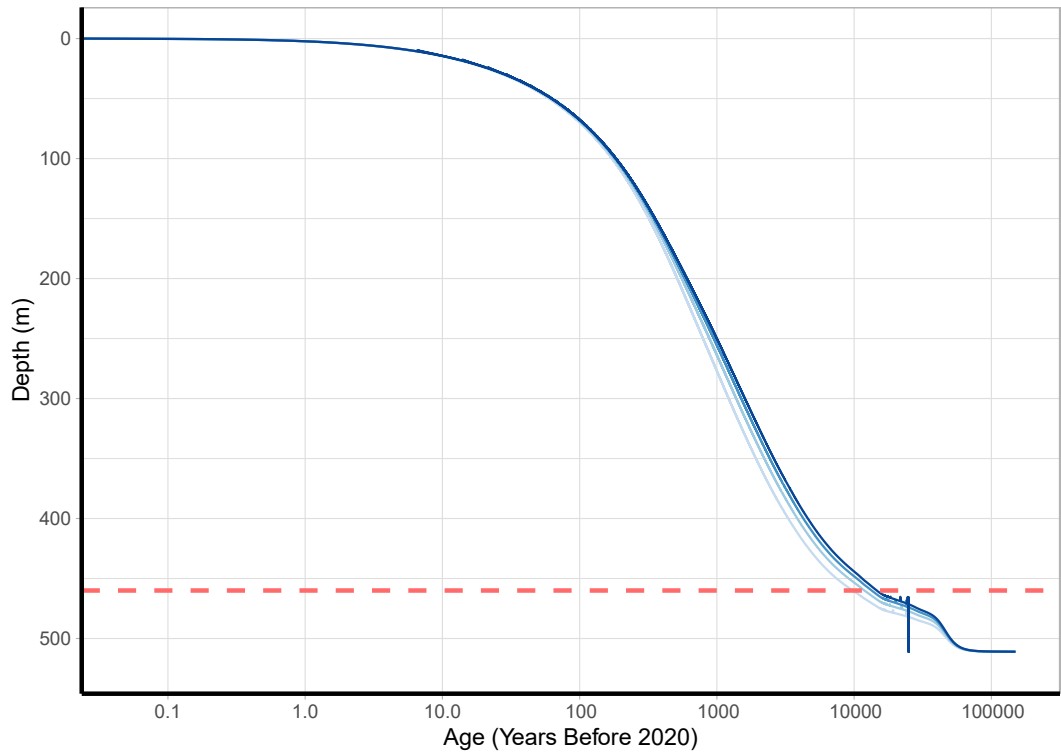

**Figure 7.** Ice thinning model predictions of ice age at the deepest point ("summit") of Sherman Island (80 m deeper than the RAID site) for a divide flow regime ($p$ = -1). The different lines are the simulations resulting from smoothed estimates of accumulation rate from age/depth modelling at the SI:RAID drill site. The simulations assume identical accumulation history based on four flank flow regimes from the SI:RAID site (p = 1 to 4, light to dark blue, Equation 3). Two maximum input age parameters were compared, 25 kyrs (dashed lines) and 150 kyrs (solid lines): they are identical until ∼20 ka. The dashed red line shows the 90 % depth (460 m) discussed in Section 4.2, used as the cut off due to unrealistic age/depth below this point, as discussed in Section 3.2.

at any chosen resolution, typically on the order of a few cm or less for continuous flow analyses of shallow to intermediate depth ice cores. Considering this limitation, the ice from Sherman Island has been dated relatively robustly from a combination of
approaches widely used in ice core analysis, with necessary accommodations made for the lower resolution samples compared with traditional ice cores. For example, annual layer counting was only possible to a relatively shallow depth (70 m, 85 years) at which point it was deliberately cut off to prevent the dubious counting of non-annual variations. In comparison, the WAIS Divide and Skytrain ice cores are annually dated to 31.2 ka BP (2850 m) and 1942 years BP (184.14 m), respectively (Sigl et al., 2016; Hoffmann et al., 2022). Furthermore, annual layer counting was aided by comparison with a short but very proximal
firn core, with higher resolution measurements; the SI:Core was sampled at 5 cm resolution compared with the average of 18 cm for the corresponding depth range of SI:RAID (21 m). The existence of such a close ice core for comparison and to assist the dating strategy is not a common occurrence in deep ice core drilling. This proximity is valuable because two such

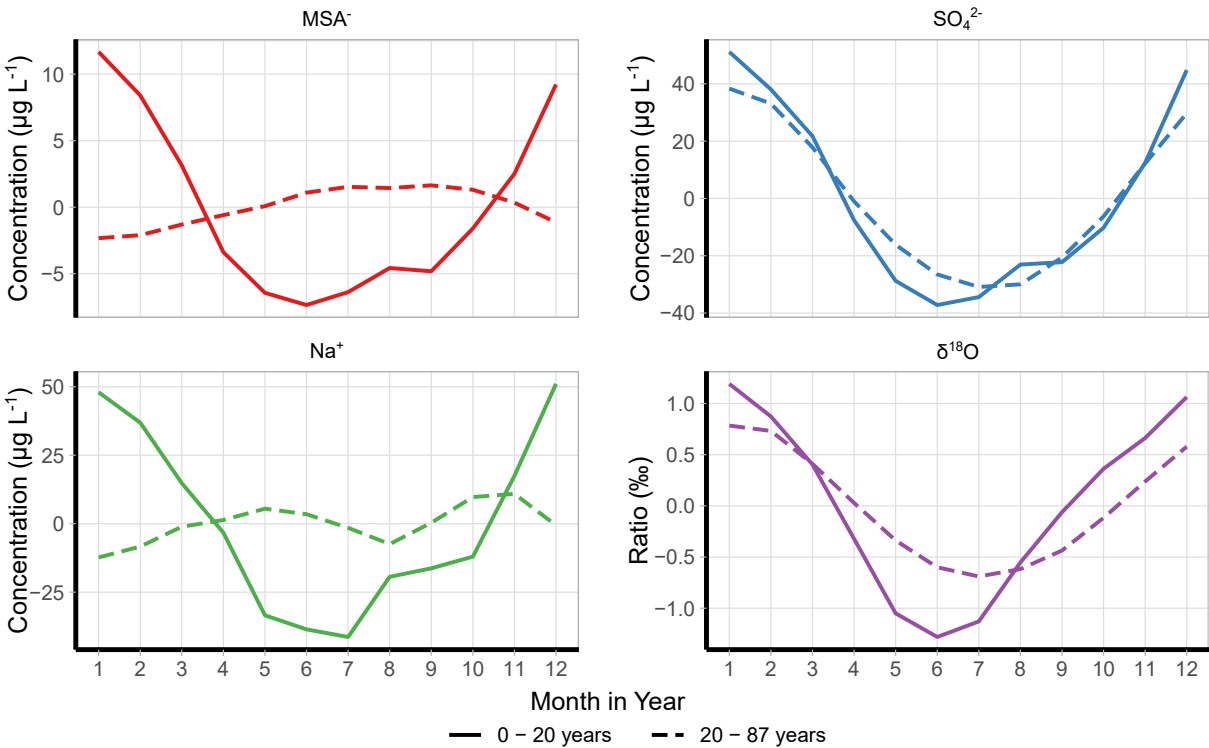

**Figure 8.** Seasonality of certain chemical species and stable water isotope composition. The x-axes show the month of the year (1 is January and 12 is December) and the y-axes show anomaly concentrations or composition of the species, relative to the annual mean. Line types are for shallow dated section (0 to 20 m) and annual layer counted section (20 to 70 m).

closely located ice cores should not have age scales which significantly deviate from each other, due to experiencing similar accumulation histories. Furthermore, despite the distance of ∼3 km between them, in the top 20 m, thinning and horizontal ice
flow are insignificant, as evidenced by the agreement of their records in Figure 2.

Volcanic synchronisation of ice core records and dating of individual ice cores using volcanic event identification using the $SO_4^{2-}$ data of cores is a standard ice core dating technique (e.g., Fujita et al. (2015); Narcisi et al. (2006); Palmer et al. (2001); Severi et al. (2017), and others). In this case, $SO_4^{2-}$ peak identification was supplemented with S isotope analysis, giving more confidence that: first, some of the $SO_4^{2-}$ peaks identified were of volcanic origin and second, that their isotopic characteristics
matched those of the expected volcanic eruption being used to assign an age horizon for its respective depth. Being confident in the eruptions identified meant that the uncertainty of those age horizons was effectively zero, being equivalent only to the estimated age increment covered by the sample depth due to the necessarily low sampling resolution.

Finally, the use of an ice thinning model that allows accumulation to vary in order to fit through empirical tie points resulted in a more realistic estimate of the age-depth relationship. This optimisation also enabled the interpolation of an age estimate
for the bottom-most sample, which would have been difficult to date in any other way, allowing an age scale for the full range

of SI:RAID samples to be developed. The model also helped to calculate the age scale uncertainty. Further use of the model is discussed below.

## 4.2 Deepest ice: Is it possible to find a continuous record beyond the Holocene in Sherman Island?

Is it possible to find a continuous record beyond the Holocene in Sherman Island? To answer that question, we discuss here the influence of local ice flow on depth/age. The ice flow near a ridge can be characterised by the proximity to the divide of flow, the vertical plane where ice starts flowing towards the opposite flanks of the ridge that is often located near the ridge (Figure 1). This is a result of the ice having non-linear rheology (Raymond, 1983). Some distance from the divide, only a few ice thicknesses, flow is dominated by shear and it is well represented by the Lliboutry approximation, with the parameter $p$ larger than 1. At the divide, however, the lower strain-rates near the bed and the non-linearity of ice translate into nearly stagnant ice. These local flow conditions have a strong influence on depth/age, as shown in Figures 5 and 7. Intriguingly, it is clear that the depth/age at SI:RAID better fits the flanking flow model and, under no reasonable assumptions of past accumulation is the forward divide flow model able to fit the observed age markers. Furthermore, the optimised age model fits the tie points well for $1 \leq p \leq 4$, demonstrating that the SI:RAID site is located at the flanks of the ice divide. However, if we were to find a site in Sherman Island at a divide of flow, we hypothesise that based on the estimated bottom-age of the ice at the SI:RAID site, the age of the ice toward the bedrock would likely reach the early Holocene (Figure 7).

The echogram from IceBridge data over the ridge near the drilling location (Figure 1, Panels B and C), shows that such flow conditions could exist on the opposite side of the ridge from our drilling site. This is because the low-strain rates near the base at the divide flow manifest conspicuously as arches in the ice structure (Vaughan et al., 1999).

Another potential drilling site on the island is near the summit, where the ice is thicker. Near the summit, the ice is approximately 80 m deeper than the SI:RAID site. Our model estimates that, assuming identical accumulation histories (taken from the modelled accumulation for $1 \leq p \leq 4$ from the SI:RAID site age/depth optimisation), under divide flow conditions ($p \approx$-1), the age at 90 % depth (459 m) is $\sim$9,700 to 16,500 years before 1950 (Figure 7). These values are dependent on the accumulation rate values given to the model, which are in turn a consequence of the assigned $p$ parameter (flow regime) in the optimisation. This is evident in Figure 7: the higher the "p-value" (in this case, the accumulation rate taken from the corresponding optimisation model with $p$ assigned between 1 and 4), the lower the accumulation rate and therefore the older the ice at the same depth. Figure 7 also demonstrates the susceptibility of the model to the maximum age input as a parameter: the outputs set to 25 ka and 150 ka follow identical age/depth relationships until approximately between 90 and 95 % depth, when rapid and unrealistic aging occurs, necessary for the model to reach its assigned maximum age. The agreement of both simulations (in terms of the maximum age supplied) until this point does, however, give more confidence to the conclusion of a Holocene ice core (or longer) being obtainable from Sherman Island.

## 4.3 Insights from the SI:RAID age scale

Bringing together our findings, we finally consider the significance of the SI:RAID age scale in a broader context. From the annually dated samples, we estimate an average modern (last 80 years) annual accumulation rate at Sherman Island of 60

cm water equivalent (standard deviation, SD, 12 cm), compared with the Regional Atmospheric Climate Model (RACMO) estimate, used for site selection, of 47 cm weq (SD 6 cm) (Mulvaney et al., 2021). From the model optimisation, allowing accumulation rate to vary at set points to enable fitting the known tie markers, the range of annual accumulation rates was between $\sim$ 46 and 110 cm weq. Measured accumulation rates at Sherman Island are thus both higher and more variable than calculated by RACMO. This is a significant finding given that efforts to reconstruct regional and continental accumulation rate histories are often heavily dependent on data assimilation techniques including the use of reanalysis products such as RACMO and ERA reanalyses (Wang et al., 2017; Thomas et al., 2017; Stenni et al., 2017). Furthermore, in the above studies, the Sherman Island region and coastal WAIS are poorly represented due to a lack of ice cores in this data-sparse region. The records from Sherman Island are therefore an important contribution to wider reconstructive efforts (e.g. Neff (2020)). The age scale presented here is robust enough to permit palaeoclimatic reconstruction using the SI:RAID data, despite its lower than average ice core resolution, which is the primary limitation of RAID records. The lack of annually resolvable cycles extending significantly beyond the reanalysis period ($\sim$40 years for this region), and the relatively low ice sheet thickness limit the length of an empirically derivable record of accumulation rate. The modelling performed for the Sherman Island summit location indicates that an ice core might not only extend back substantially further in time (thousands of years), but would be more highly resolvable. Further investigation of the simulations indicates the potential for seasonally resolvable variations to 0.7 to 1ka, with annual resolution to at least 2ka, assuming an analytical resolution of 5 cm or greater. By 90 % depth, annual layers are approximately 3 mm thick.

## 5 Conclusions

An age scale for the SI:RAID samples, which extends back to more than 1240 years before present day and is currently the longest ice core from the coastal West Antarctic and western Antarctic Peninsula regions, has been presented. The use of chemistry data and S isotope measurements on RAID-drilled ice to establish an age scale is presented here for the first time, in addition to stable water isotopic measurements. The SI:RAID data will make a valuable contribution to regional, continental and global compilation projects such as PAGES-2k (PAGES 2k Coordinators, 2017). There is a lack of a recent Antarctic Holocene ice core composite, and the 1000 years of climate data from Sherman Island could make a significant contribution to work such as this (Masson et al., 2000). If in the future a full-scale drilling campaign on Sherman Island were to be carried out, a longer, full Holocene record for this region in West Antarctica could probably be obtained by drilling to bedrock at the deepest point of Sherman Island. Such a record would be critical in gaining insights into West Antarctic interglacial variability. The records contained in the existing Sherman Island data have the potential to constrain climate history over the last millennium in this important and vulnerable region of West Antarctica and represent a valuable addition to the ice core community.

*Author contributions.* The manuscript was written by IR with contributions from CM, HP, EW, DT and JL. The RAID ice was drilled and sampled by RM, IR and DT. The Sherman Island firn core was drilled by DT. The RAID chemistry samples were analysed by IR, and the

380 isotopes by IR and RM. The Sherman Island firn core was processed and analysed and data made available for use by DT. The sulfur isotope analysis was done by HP and ED. The annual layer counting was done by IR, with ice core contributions from DT. The volcanic horizon identification was done by IR, HP and EW. The ice models were developed by CM and used by IR with assistance from CM. The age scale uncertainty estimation was done by IR with assistance from EW and CM. The seasonality analysis was done by IR. Age predictions were done by IR, CM and EW. The radargram data were processed by HM and JL.

*Competing interests.* The authors wish to declare that Eric Wolff, one of the co-authors of this manuscript, is a member of the editorial board for Climate of the Past.

*Acknowledgements.* This project has received funding from the European Research Council under the Horizon 2020 research and innovation programme (grant agreement No 742224, WACSWAIN). This material reflects only the author's views and the Commission is not liable for any use that may be made of the information contained therein. EW has also been funded through a Royal Society Professorship.

The drilling of the Sherman Island shallow ice core was supported by the Collaborative Antarctic Science Scheme (CASS-168). Data obtained from the Sherman Island shallow ice core was funded by a CONICYT–Becas Chile and Cambridge Trust fellowship awarded to Dieter Tetzner, grant number 72180432.

The authors would like to thank Pete Akers, Eric Steig and a third, anonymous reviewer, whose thoughtful consideration of this paper contributed to its improvement.

Finally we thank Shaun Miller and Jack Humby for their assistance with the lab analyses necessary for this project.

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
