# Peer review of "An age scale for new climate records from Sherman Island, West Antarctica"

_Climate of the Past, 2022_

## Author Comment (AC1)

**Overview:**

The authors here present their chronology development for a borehole from Sherman Island in West Antarctica. They use multiple methods to determine their chronology, largely based on the constraints of their sampling and changes in the ice chemistry resolution with depth. They find their borehole covers back ~1200 years and propose that ice near the bedrock could be early Holocene.

Overall, I find that their methodology is sound (although I will admit that I do not have personal experience with flow or thinning modelling). It generally accomplishes its goal of presenting an age-depth scale for the location, but broader discussion is rather limited outside of extrapolating what the bedrock age of the ice might be. The writing is generally good and easy to follow, with only a few minor technical/grammar issues highlighted in the technical comments.

As a stand-alone manuscript, this comes across lacking in parts at times. This manuscript is clearly complementary to Rowell et al, 2022, but readers unfamiliar with that paper will feel rather lost reading the present manuscript. I personally didn't understand many of the mechanics and specifics alluded to in this manuscript until I had to go read Rowell 2022. While it is of course fine to point out that the details on certain methods and analyses are in Rowell 2022, this manuscript in review needs to be able to stand alone enough that reader can understand all the basic results and discussion in the manuscript.

Following on that, the overall discussion of the final chronology and work feels a bit underwhelming and undeveloped. I don't think that a massive expansion of discussion is required (or warranted), but it would be nice to see some broader impacts and discussion about the chronology results. Is the final accumulation rate different from what is observed elsewhere or expected from models? How does the quality/resolution compare to the SI Core or other core-based results from similar sites? Are there any broad lessons learned about in how this method could be applied or where it should be applied elsewhere? Note that these aren't questions I'm requiring the authors to answer; they are simply pointing out some ways that a deeper discussion could make the manuscript more impactful than just largely a technical report specific only to this Sherman Island site.

Altogether, the manuscript is generally a solid report on the results of a chronology production. Should the editor(s) desire more than this, I think the paper could be made more impactful by developing the discussion more (both what exists on the bedrock age modelling and adding some broader comparative context and/or applicable lessons).

*We thank you for your thoughtful consideration of this paper and for your assessment that the methodology and conclusions we present are sound. Upon reflection it is clear that there was too great a need for referral to the Rowell 2022 paper for this manuscript*

*to be considered a standalone text. And further discussion, as suggested, would enhance the paper and give more impact.*

*We have added further explanation about the chosen drill site (why and how it was selected) and use of the RAID to the introduction as requested in your following comments. We have also elaborated in the discussion and addressed some of the questions you raise in your review. As a result of other reviewers' comments we are re-visiting some of the modelling work and re-running the model to use the WAIS Divide accumulation rate as the reference for the record, as you also mention this in a later comment. Furthermore we have recently obtained a radargram from the Icebridge radar data available for Sherman Island and would like this to be included in the revised paper because it will enable us to expand on the discussion of bedrock age and ultimately the potential for a future ice core from this site.*

**Major Comments:**

Introduction: The first paragraph sets the scene well, but I struggled to understand exactly the point and details of the project in the next two paragraphs. For example, is the RAID system different from other drilling systems? Why was Sherman island chosen? It is serving as a constraint of what? Moreover, what is the actual point and objective of THIS paper? There doesn't need to be a ton of detail on this stage, but more clarity and structure to prepare the reader for what they are about to read.
*Thank you for explaining clearly what is lacking from the introduction. Instead of depending on referral to other papers (e.g. Mulvaney 2021 and Rowell 2022) for details about the project background, we have elaborated more on this in the introduction (now lines ~34-38). Specifically, we have explained that LIG ice from Sherman island would constrain the LIG WAIS stability in an additional location (if present) and stable water isotope records could indicate temperature and elevation history at the site as well as providing further palaeoclimate records (chemistry).*

35: The use and reasoning of the RAID aren't clear here. Since it isn't stated what the RAID is, or how it differs from normal drilling, I don't know why it would be chosen in light of the risk of Sherman Island. (Reviewer note: Later I see that Rowell 2022 has this information, but the basics need to be summarized here in this manuscript also).
*To avoid unnecessary referral to other papers, we have added a description of the RAID and how it differs from conventional drilling (now lines 45-50).*

35: Why was Sherman Island chosen, then, if it had such high risk of not contributing to the goals of the WACSWAIN project? Did it have other virtues that warranted the risk, or was is simply logistically easy?
*We have altered the introduction (between lines ~34-55) to address this concern, including addressing the comments above and trying to stress that the use of the RAID was **because** of the risk that Sherman Island presents - of not containing old ice - and that a useful and long (potentially LGM) "core" could still be obtained, and still from a new and interesting location in the WAIS.*

Drilling and measurements: Some more information is needed here. Although papers with detailed descriptions are cited, this manuscript should include the minimum details required to understand the rest of the paper without having to look up a second paper. Namely, what sort of samples is the RAID bringing up? If they are chips, are they in stratigraphic order? What is modern environment at the drill site (accumulation, etc)? What is the vertical resolution of samples if they are mixed? These are all critical to understanding the rest of the paper, and a reader shouldn't have to go find a second paper to get this information.

*Thank you, you are right to suggest that we expand more here rather than require readers to look elsewhere. We have added more detail about the samples, specifically, a description of how the ice chippings were sampled, their resolution, stratigraphy and mixing, and some more details about Sherman Island as a drill site (e.g modern accumulation rate originally estimated from RACMO to be 46.8 cm, but ice core analysis of last 20 years suggests higher rate of 58.3 cm).*

Figure 1. A close up map of the Sherman Island region and drill site would be beneficial to understanding the local geography and ice structure/flow.

*Yes we agree, and this will be included in a revised paper along with some other details to the map that we think would be beneficial (e.g. pointing out the relative location of the Thwaites and Pine Island glaciers referred to in the text, and other reference points such as existing WAIS ice cores including WAIS Divide).*

105: Why was EPICA Dome C chosen as the assumed proportional accumulation as opposed to, say, WAIS? Is there any evidence to think that Dome C and SI would be proportional? They are very different climate systems.

*This is a fair question and we are re-visiting the modelling for this paper to address this and similar comments by other reviewers. We would like to use WAIS Divide as a reference accumulation record, in addition to the modelling already done. We expect that this will not significantly alter the conclusions or final age scale for two reasons: 1) the accumulation rate of both sites (WAIS Divide and Dome C) has not significantly changed over the last ~1000 years and 2) the model smooths the accumulation record. However you are right to point this out and we should test this thoroughly.*

110: This sentences reads awkwardly. Better fitting than the markers? Better fitting judged in what way? "It became clear" gives us no insight into what decisions were made, statistics performed, or observations made.

*At the request of another reviewer, this sentence has been moved and, at your request, rephrased to make the approach more clear to the reader. We explain that the annual layer counting markers (shown in Figure 5) are more closely matched to the flank-flow depth/age model than the divide-flow, providing justification for using this depth/age model as the basis for identifying volcanic markers deeper in the core.*

113: Again, if it is clear, you don't need to state it to the reader. You should present what made it clear.

*Thank you, this is a good point. We have been more specific in explaining that with an average annual layer thickness of 87 cm in the top 70 m, a sample resolution of 19 cm*

*throughout the borehole would mean sample resolution remains annual or greater until at least 250 m depth (through comparison with the depth/age from the flank-flow model), enabling resolution of individual volcanic events in the sulfate record.*

114: Annual resolution isn't necessarily required to detect volcanic events. They are found at Dome C, for example, and the signal there is blurred over annual levels. In fact this raises an interesting question: since the sulfate fallout can extend over several months, did any of the volcanic peaks interfere with your annual dating by SO4 peaks?
*This is an important consideration that you are right to point out. However, we deliberately stopped annual layer counting at 70 m depth, and the shallowest volcanic events that we identified were the 1815 and 1809 eruptions at 125-130 m depth. We demonstrate in Figure 4 that these eruptions do indeed appear over several RAID samples. To address your main concern, we know that this does not interfere with the annual layer counts here, however if in future use of the RAID if annual layer counting was carried out to deeper depths, this would need to be considered.*

140: Note that Kuwae is still disputed as a source for the 1458 eruption (https://doi.org/10.1038%2Fs41598-019-50939-x). Should become more clear with some research over the next decade, but you could add a "commonly attributed to Kuwae" or "eruption in 1458 (possibly Kuwae)" style to be safe.
*Thank you for this helpful comment, we have made your suggested edit to the text. We have also edited line 163 (now line 180) where Kuwae is mentioned again and corrected a typo for the date at this line from 1485 to 1458.*

Figure 4: The legend is hard to see with its small size and placement. I recommend enlarging it and placing it at the top and/or direct labelling the y-axes and lines.
*Thank you, you are right that the legend is small. In addition to some other changes to the figures (colour scheme and line thickness), we will adjust the legend to make it more clear.*

233: What is known about the bed of the glacier here? Is it thought to have significant melt? Is the assumption about no basal melting simply for calculations, or is there geologic evidence supporting this assumption?
*This is an important question and we will add more detail to the discussion here using the data we have available. Specifically, we have borehole temperature measurements to 323 m which extrapolating to the bed give a basal temperature of -6 C (Mulvaney 2021). This is worth reiterating at this point in the discussion and we will make these changes.*

253: Again, there are references to the uniqueness or special nature of the RAID samples, but they were never described in this manuscript.
*Thank you for pointing this out. We have re-phrased this section to remove the vague description of the "unique RAID samples". We have been more clear about the accommodations that were necessary - cutting off the annual layer counting at 70 m and comparing with a nearby and higher resolution firn core.*

Discussion: Sections 5.1-5.3 are very short and do not add much discussion of the previous parts. I'd argue that they could simply be appended to their appropriate section in the results. Section 5.4 has some intriguing points, but is underdeveloped and feels a bit like an afterthought.

*Thank you for your thoughts on the discussion. I agree that we could do more to develop this part of the paper. As a result of reviewer comments, we will be re-visiting the modelling (e.g. using WAIS divide as the reference accumulation rate record) and have obtained some recently processed radar data from the Icebridge project from flyovers on the island which we would like to add as further evidence for the conclusions we make in section 5.4. Along with these additions and changes to the discussion we will consider re-structuring the earlier discussion sections as you suggest.*

Data availability: Are the data ($d_2H$, chemical species, etc) that led to the creation of the age model in an online archive or otherwise available? I didn't see a link at Rowell 2022 (unless they are contained in that paper's supplementary material).

*You are right to point this out - the data are not currently available. We would like to make the data for the full records available and will submit them to Pangaea. However we are also in the process of writing papers which will present the full, dated, stable water isotope and chemistry records and would like to publish them as standalone datasets when these papers are published, rather than there also be an additional dataset of just the top ~70 m of data, as we think this would be confusing and not necessary. So one option would be for us to place the data on Pangaea with an embargo until these publications are available. Another possibility is of course to submit the data presented here as an additional supplement to this paper. I hope this makes sense and that either of these options seems reasonable.*

**Technical comments:**

17: The acronym MISI is never used again in the manuscript, and therefore it is not necessary to define it here.
*The acronym has been removed.*

21: Both of these acronyms are only used once elsewhere. Remove acronyms and just spell out in second instance.
*The acronyms have been removed.*

26: LIG is used three other times. It may be better for readability to simply spell it out each time.
*Due to expansion of the introduction and discussion, LIG is now used more frequently throughout the paper, so the acronym has been kept in line 26.*

45: Suggestion to consider flipping the sentences in this section so that you give a brief summary of the important necessary information here, and then end with the sentence that "a detailed description…".
*We have made other changes to this section as per your previous suggestion, so this is no longer needed.*

Section 3: The numbering scheme is odd with 3.0.1, 3.0.2, etc. Since there is no 3.1, shouldn't these simply be sections 3.1, 3.2, etc.?
*Yes thank you for pointing this out, the numbering has been corrected.*

67: Is SI core the name of the ice core, or an abbreviation for Sherman Island? If it is the core name, perhaps adding a date and/or length to the end will make it more unique of an identifier. If it is an abbreviation, Sherman Island isn't abbreviated elsewhere (which I think is good).
*Yes, SI:Core is the name we use to denote the 20 m firn core, to separate it from SI:RAID, which is named as such to make it clear we are talking about a "core" of RAID chippings, not a solid ice core. We name the RAID records from Little Dome C in this manner too (LDC:RAID1 and LDC:RAID2 for the two RAID boreholes which exist). For consistency and clarity over the type of ice we are referring to, we would like to maintain this naming system. You are right to point out that this is the first introduction of the naming convention in this paper, however, so we have introduced "SI:RAID" earlier in the text (now line 71) and clarified the naming of the firn core at line 83 (previously 67).*

73: This phrasing seems odd to me. Perhaps "This variability could reflect the local geography of Sherman Island…"
*Agreed, upon reflection the intended meaning was not clear, we have taken your suggestion.*

74: "and IS then closely by…"
*Corrected.*

Figure 2 (and others): The line thicknesses are rather thick, which makes it hard to see small details where exactly indicated lines are falling. Consider making the thicknesses thinner.
*Yes, we will reduce the thickness of the lines in Figures 2, 3 and 4.*

102: Sentence starting with "That is" is a fragment.
*Corrected.*

111: a priori not prior, I think here.
*Due to another reviewer's comment, this sentence has been moved and re-worded so this phrase is no longer needed.*

Figure 4 (and others): This particular shade of red and green is difficult for many colorblind people. The colors aren't overlapping, so it isn't as big of a concern, but something to consider if you revise your figures. However, someone colorblind would not be able to make the connection from your legend to the lines.
*Thank you for pointing this out. We will change the colour schemes used in the figures throughout the paper and be sure to check the figures again using a colour blind simulation tool.*

178: This paragraph could be merged with previous.
*Agreed, we have merged the first three paragraphs of this section into one.*

Fig 7: Is this the best x-axis display for this plot? It seems to add more confusion and oddness since the yr BP axis goes negative and positive on the log axis.

*Thank you for this comment. Presenting the age as relative to 1950 necessitates presenting negative values, and due to the desire to present this modelled data on a log x-axis scale to be able to visualise the range in the modelled outputs at an appropriate scale, this meant adding the negative values on the x-axis to -100. However, you are right that this could be confusing, so we will revisit the scale used here and either present the data as relative to 2020 (meaning no negative values) or re-consider the use of the log scale entirely. As there is more work to be done on the modelling as suggested by other reviewers, we will endeavour to address all the comments as closely as possible.*

---

## Author Comment (AC3)

**General comments**
This is a great paper, and the authors are to be commended!   They have written the first paper to convincingly demonstrate the utility of the RAID system.  They have also provided a really impressive and careful analysis of the timescale for the Sherman Island site, which will prove useful as a climate record, and strongly supports the idea of drilling a complete "normal" ice core record at this site.
*Thank you for your kind words and your assessment that both the RAID system is convincing and that the age scale for the Sherman Island RAID data is adequate. I am happy to hear that you agree Sherman Island would make a good future drill site!*

**Technical comments**
I have a few criticisms that I suggest should be considered in a revised paper.

First, a few minor grammar/style things:

Please define abbreviations before using them.  In the asbstract, "LIG" is used, but not defined.
*LIG abbreviation now defined in abstract, and additional missing definitions elsewhere in the text have been identified and defined*

The terms SI:RAID and SI:CORE are used but not defined. At first, I thought SI might mean Supplementary Information!
*Thank you for pointing this out, both terms (SI:RAID and SI:Core) have now been defined in the text when they are introduced. I will steer clear of any use of "SI" on its own for this reason.*

In the abstract, it is stated that the new record is the "oldest, continuous, ice-derived palaeoclimate records for the coastal Amundsen-Bellingshausen Sea sectors."  This is a stretch.  WAIS Divide gets its snow accumulation from the Amundsen Bellinghausen Seas, and probably reflects the Amundsen-Bellingshausen Sea sectors almost as well as the Sherman Island site does.  Instead, I would say "oldest, continuous, ice-derived palaeoclimate records *IN* the coastal Amundsen-Bellingshausen Sea sector.
*Thank you, you are right that is an important clarification. We have made the change you suggest.*

On Page 1, where Joughin and Alley 2011 are cited, I suggest a more primary source, rather than a review paper.
*Thank you, we have provided a reference instead to Joughin et al, 2014*
*https://doi.org/10.1126/science.1249055*

Similarly, on page 2, I don't think Deconto and Pollard is the best citation for the statement that the WAIS may have collapsed in the past.  It would be better to cite evidence for this from sea level records or other papers that provide data suggestive of this.  Deconto and Pollard is a modeling paper that *assumes* WAIS has collapsed.  It is not a source of evidence that it did.
*Thank you, this is a good point. The Deconto and Pollard paper is useful for helping to*

*set the context of why investigating the WAIS during the LIG further using empirical (ice core) evidence is important, but it is not the correct citation for the statement here. I have instead referred to Dutton et al 2015. I have amended the subsequent sentence to "WACSWAIN… aims to use ice core records to investigate the WAIS during the LIG, to supplement existing modelling studies (e.g. Deconto and Pollard)"*

**Major comments**

On page 4, the model of Martín et al. (2015) is briefly described. But I would like to understand the motivation for using this model, rather than simply models such as that of Dansgaard-Johnsen, which is commonly used.  Furthermore, elsewhere in the paper other models (e.g. Liboutry) are used, so this is confusing.  Also, it is stated that horizontal advection is neglected, but nothing is said about whether this is a reasonable assumption for this site.

*Thank you for your helpful comment. As I explain below, we have some work to do with re-visiting the model simulations in general and improving the explanation of the modelling in the text, so thank you for pointing out this confusion specifically. We will address this and provide further description and explanation for the use of this model.*

On page 5, I find the Caption for the graph to be confusing.  It is stated that divide flow is shown in red, flank flow in blue, and that green shows the optimized model.   This implies that the best model is inconsistent with both divide flow and flank flow.  But in the text, it's clear that this is NOT the difference between the curves.  The difference is that assumptions made about the snow accumulation rate were probably too simple.  That's a nice result, but it is not at all obvious.  As far as I can tell, the green curve actually uses divide flow, but allows for variable accumulation.  (If I am wrong, then I am confused by the text, and some work is needed to make it clear what has been done).  I strongly suggest renaming the curves, with something like "divide flow with Dome C-proportional accumulation", "flank flow with Dome C-proportional" and "divide flow (??) and optimized accumulation".

*I assume this refers to Figure 5? I agree that the explanation in the text needs some more work to improve the clarity. We are currently working to improve the model simulations (partly as per your below suggestion and as a result of the other reviewers' comments and subsequent discussion), and will address this as part of the revised manuscript. Thank you very much for your clear description of what is confusing about the figure and the accompanying text.*

I have one significant criticism.  Why is Dome C used for accumulation rate in the first place?  It is about as far from Sherman Island as possible.  Why not use WAIS Divide?  Even South Pole would be better!   The authors show clearly that the assumption that the accumulation history is proportional to that at Dome C does not work -- if it did, then the "optimized" timescale would be very similar to the "divide" timescales.  There is a missed opportunity here.  The authors do not show what accumulation history goes with the optimized timescale (they should show that!).   It would be very interesting to know whether that history agrees better with Dome C or with WAIS Divide or South Pole.

I would very much like to see the "divide" and "flank" timescales re-calculated, using WAIS Divide (and perhaps also South Pole), and some commentary on which one provides the closest fit to the optimized timescale.

*Thank you for this assessment. I will address these two comments together as they are concerning the same criticism. Another reviewer also raised concerns about the use of the Dome C accumulation rate as the reference. We will do another set of model simulations using the WAIS-Divide accumulation rate record. We agree that this is an important comparison to make because the WAIS Divide and Dome C climates are so different. We don't expect that this will significantly alter the results, because at both sites for the last ~1000 years (the approximate age range covered by the Sherman Island RAID samples) there has not been a large change in accumulation rate, and furthermore the model smooths the accumulation record. However, this should also be made more clear in the text, and is a parameter which in turn we could also investigate further. You are correct that we need to be more clear in the text in our explanation of the model runs presented and we thank you for your constructive comments.*

---

## Author Response (AR1)

**Introduction and general comments**

The individual responses to reviewers (below) in this document are based on the previous responses, updated to reflect the work that has been done since the original submission. We also provide a general overview of the changes made, and their significance, because there were common themes in the reviewers' comments which necessitated some significant changes to the manuscript. For clarity we are emphasising here the main changes made.

**Overview of the major changes made to the manuscript**
- We have re-visited the age/depth model optimisation: this is the biggest change and is described in more detail below and in the individual responses to the reviewers.
- We have included the radar echogram of layers in the ice sheet at Sherman Island and subsequent discussion.
- We have added an extra section to the discussion, and changed some of the section structure of the paper.
- We have removed the Appendix Table A1, because the data in this table are all provided in a more comprehensive supplement (of S isotope data).
- We have added an additional supplemental data set of the annual layer counting.
- We are in the process of publishing the whole (dated) SI:RAID dataset on Pangaea. The data have been uploaded and are currently with a Data Editor in Pangaea data quality control.

**Modelling work re-visited**

Two reviewers suggested that the modelling of age/depth relationship at Sherman Island would be better served by using the accumulation rate record from WAIS Divide as a reference record, rather than from EPICA Dome C. In revisiting the modelling work for this purpose, we discovered that the age/depth optimisation had been incorrectly carried out, without appropriate time points given for changes in accumulation rate between known age/depth tie points (annual layers and volcanic markers). This work has been completely re-done with these two changes: using WAIS Divide as the accumulation rate reference, and with including fixed time points at which the accumulation rate is allowed to vary, in order to best fit the known age/depth tie points.

We would like to draw the reviewers' attention to changes in the text in Section 2.2.2 and 2.2.4, as well as the different output in Figures 5 to 7. We are open to further discussion on this, but our understanding is that regarding the dating of the existing ice samples, the changed results are not so much the result of using WAIS Divide as the reference (although the reviewers are correct to suggest this), but rather of including time points at which the model enables stepwise changes in accumulation rate. It is clear (for example in Figure 5), that a shift in accumulation rate is necessary to fit the 1815 and 1809 volcanic events and annual layer counts as well as the deeper volcanoes. There is still uncertainty about the timing of this change in accumulation rate, which we discuss in the text. However, the result that the modelled output fits well with the age markers, enables us to use the modelled age/depth in the age scale interpolation, rather than using a statistical approach (which was the case in the original submission). We therefore conclude that the changes to the modelling have increased our confidence in the age scale.

Regarding the use of the WAIS Divide reference accumulation record, this becomes more important for modelling beyond where tie points exist. This is because, as we describe in section 2.2.2 and 2.2.4, the piecewise linear function for accumulation rate history, $a(t)$, is

optimised to fit the data, and beyond the last tie point the accumulation rate is assumed relative to WAIS Divide. The results for the predictions of deeper sites have somewhat changed, although they do not change the key findings of the paper, as we describe below.

**Significance of changes**

In terms of the main findings of the paper, the conclusions have not substantially changed since the original submission, despite the amount of new modelling work carried out. For example, the SI:RAID age scale of the ice obtained, has changed from 1082 years before 1950, to 1176 years before 1950. Thus the new age scale has the bottom-most ice dated as 94 years older. Because of this change, and as a result of changes to the modelling described in the text, we assign a slightly larger uncertainty to the entirety of the new age scale. For the predictions of depth/age at the deepest site on Sherman Island, a larger change has resulted from the updated modelling, but now with more consistent findings between the model simulations. The age at 90% depth has changed from a range in the original manuscript of 10.7 to 23.6 ka, to an updated estimate of 9.7 to 16.5 ka. In either case, the main conclusion that there is likely a potential full Holocene ice core at Sherman Island remains valid.

**Overview:**

The authors here present their chronology development for a borehole from Sherman Island in West Antarctica. They use multiple methods to determine their chronology, largely based on the constraints of their sampling and changes in the ice chemistry resolution with depth. They find their borehole covers back ~1200 years and propose that ice near the bedrock could be early Holocene.

Overall, I find that their methodology is sound (although I will admit that I do not have personal experience with flow or thinning modelling). It generally accomplishes its goal of presenting an age-depth scale for the location, but broader discussion is rather limited outside of extrapolating what the bedrock age of the ice might be. The writing is generally good and easy to follow, with only a few minor technical/grammar issues highlighted in the technical comments.

As a stand-alone manuscript, this comes across lacking in parts at times. This manuscript is clearly complementary to Rowell et al, 2022, but readers unfamiliar with that paper will feel rather lost reading the present manuscript. I personally didn't understand many of the mechanics and specifics alluded to in this manuscript until I had to go read Rowell 2022. While it is of course fine to point out that the details on certain methods and analyses are in Rowell 2022, this manuscript in review needs to be able to stand alone enough that reader can understand all the basic results and discussion in the manuscript.

Following on that, the overall discussion of the final chronology and work feels a bit underwhelming and undeveloped. I don't think that a massive expansion of discussion is required (or warranted), but it would be nice to see some broader impacts and discussion about the chronology results. Is the final accumulation rate different from what is observed elsewhere or expected from models? How does the quality/resolution compare to the SI Core or other core-based results from similar sites? Are there any broad lessons learned about in how this method could be applied or where it should be applied elsewhere? Note that these aren't questions I'm requiring the authors to answer; they are simply pointing out some ways that a deeper discussion could make the manuscript more impactful than just largely a technical report specific only to this Sherman Island site.

Altogether, the manuscript is generally a solid report on the results of a chronology production. Should the editor(s) desire more than this, I think the paper could be made more impactful by developing the discussion more (both what exists on the bedrock age modelling and adding some broader comparative context and/or applicable lessons).

*We thank you for your thoughtful consideration of this paper and for your assessment that the methodology and conclusions we present are sound. Upon reflection it is clear that there was too great a need for referral to the Rowell 2022 paper for this manuscript to be considered a standalone text. And further discussion, as suggested, would enhance the paper and give more impact.*

*We have added further explanation about the chosen drill site (why and how it was selected) and use of the RAID to the introduction as requested in your following comments (lines*

*34-56). We have also elaborated in the discussion and addressed some of the questions you raise in your review. We revisited the modelling work using WAIS Divide as the reference accumulation rate record, which you mention in a later comment. Furthermore we have collaborated with two new co-authors to present the radar echogram from IceBridge radar data available from Sherman Island.*

**Major Comments:**

Introduction: The first paragraph sets the scene well, but I struggled to understand exactly the point and details of the project in the next two paragraphs. For example, is the RAID system different from other drilling systems? Why was Sherman island chosen? It is serving as a constraint of what? Moreover, what is the actual point and objective of THIS paper? There doesn't need to be a ton of detail on this stage, but more clarity and structure to prepare the reader for what they are about to read.
*Thank you for explaining clearly what is lacking from the introduction. Instead of depending on referral to other papers (e.g. Mulvaney 2021 and Rowell 2022) for details about the project background, we have elaborated more on this in the introduction (now lines ~34-57). Specifically, we have explained that LIG ice from Sherman island would constrain the LIG WAIS stability in an additional location (if present) and stable water isotope records could indicate temperature and elevation history at the site as well as providing further palaeoclimate records (chemistry).*

35: The use and reasoning of the RAID aren't clear here. Since it isn't stated what the RAID is, or how it differs from normal drilling, I don't know why it would be chosen in light of the risk of Sherman Island. (Reviewer note: Later I see that Rowell 2022 has this information, but the basics need to be summarized here in this manuscript also).
*To avoid unnecessary referral to other papers, we have added a description of the RAID and how it differs from conventional drilling (now lines 46-56).*

35: Why was Sherman Island chosen, then, if it had such high risk of not contributing to the goals of the WACSWAIN project? Did it have other virtues that warranted the risk, or was is simply logistically easy?
*We have altered the introduction (between lines ~40-47) to address this concern, including addressing the comments above and trying to stress that the use of the RAID was* **because** *of the risk that Sherman Island presents - of not containing old ice - and that a useful and long (potentially LGM) "core" could still be obtained, and still from a new and interesting location in the WAIS.*

Drilling and measurements: Some more information is needed here. Although papers with detailed descriptions are cited, this manuscript should include the minimum details required to understand the rest of the paper without having to look up a second paper. Namely, what sort of samples is the RAID bringing up? If they are chips, are they in stratigraphic order? What is modern environment at the drill site (accumulation, etc)? What is the vertical resolution of samples if they are mixed? These are all critical to understanding the rest of the paper, and a reader shouldn't have to go find a second paper to get this information.
*Thank you, you are right to suggest that we expand more here rather than require readers to look elsewhere. We have added more detail about the samples, specifically, a description of*

*how the ice chippings were sampled, their resolution, stratigraphy and mixing (lines 53-60), and some more discussion about Sherman Island as a drill site (e.g modern accumulation rate originally estimated from RACMO to be 47 cm, but ice core analysis of last 20 years suggests higher rate of ~60 cm), a new section in the discussion, lines 344-361.*

Figure 1. A close up map of the Sherman Island region and drill site would be beneficial to understanding the local geography and ice structure/flow.
*Yes we agree. We have included an updated regional map (Panel A) which includes other ice core sites for reference, and a second panel showing Sherman Island close up, with the IceBridge flyover lines, and drill sites on the island. We also show the radar echogram in this figure as Panel C, because it leads on from the IceBridge lines shown in Panel B.*

105: Why was EPICA Dome C chosen as the assumed proportional accumulation as opposed to, say, WAIS? Is there any evidence to think that Dome C and SI would be proportional? They are very different climate systems.
*This is a fair question and we have re-visited the modelling for this paper to address this and similar comments by other reviewers. We have used WAIS Divide as a reference accumulation record and corrected the age/depth optimisation using time points at which the accumulation rate can vary in order to fit the tie points. Using WAIS Divide is more appropriate, but only impacts the results of the predictions of deeper age/depths. For the depth range over which tie points exist, the modelled accumulation rate history is dependent on the age/depth of the tie points, not on the WAIS Divide accumulation rate.*

110: This sentences reads awkwardly. Better fitting than the markers? Better fitting judged in what way? "It became clear" gives us no insight into what decisions were made, statistics performed, or observations made.
*At the request of another reviewer, this sentence has been moved and, at your request, rephrased to make the approach more clear to the reader. We explain that the annual layer counting markers (shown in Figure 5) are more closely matched to the flank-flow depth/age model than the divide-flow, providing justification for using this depth/age model as the basis for identifying volcanic markers deeper in the core.*

113: Again, if it is clear, you don't need to state it to the reader. You should present what made it clear.
*Thank you, this is a good point. We have been more specific in explaining that with an average annual layer thickness of 87 cm in the top 70 m, a sample resolution of 19 cm throughout the borehole would mean sample resolution remains annual or greater until at least 250 m depth (through comparison with the depth/age from the flank-flow model), enabling resolution of individual volcanic events in the sulfate record.*

114: Annual resolution isn't necessarily required to detect volcanic events. They are found at Dome C, for example, and the signal there is blurred over annual levels. In fact this raises an interesting question: since the sulfate fallout can extend over several months, did any of the volcanic peaks interfere with your annual dating by SO4 peaks?
*This is an important consideration that you are right to point out. However, we deliberately stopped annual layer counting at 70 m depth, and the shallowest volcanic events that we identified were the 1815 and 1809 eruptions at 125-130 m depth. We demonstrate in Figure 4 that these eruptions do indeed appear over several RAID samples. To address your main*

*concern, we know that this does not interfere with the annual layer counts here, however if in future use of the RAID if annual layer counting was carried out to deeper depths, this would need to be considered.*

140: Note that Kuwae is still disputed as a source for the 1458 eruption (https://doi.org/10.1038%2Fs41598-019-50939-x). Should become more clear with some research over the next decade, but you could add a "commonly attributed to Kuwae" or "eruption in 1458 (possibly Kuwae)" style to be safe.
*Thank you for this helpful comment, we have made your suggested edit to the text. We have also edited line 163 (now line 191) where Kuwae is mentioned again and corrected a typo for the date at this line from 1485 to 1458.*

Figure 4: The legend is hard to see with its small size and placement. I recommend enlarging it and placing it at the top and/or direct labelling the y-axes and lines.
*We have made several alterations to this figure: the S isotope marker lines are thinner to be more clear, the lines of the chemical data are thinner, and the legend has been removed and labels added to each line instead. This also solves the problem of the colour theme, because it is clear which line is which regardless of their colour.*

233: What is known about the bed of the glacier here? Is it thought to have significant melt? Is the assumption about no basal melting simply for calculations, or is there geologic evidence supporting this assumption?
*This is an important question and we will add more detail to the discussion here using the data we have available. Specifically, we have borehole temperature measurements to 323 m which extrapolating to the bed give a basal temperature of -6 C (Mulvaney 2021). We have added this point to now line 120 in the description of the model equations.*

253: Again, there are references to the uniqueness or special nature of the RAID samples, but they were never described in this manuscript.
*Thank you for pointing this out. We have re-phrased this section to remove the vague description of the "unique RAID samples". We have been more clear about the accommodations that were necessary - cutting off the annual layer counting at 70 m and comparing with a nearby and higher resolution firn core. With the changes made to the introduction which describe the nature of the RAID samples in more detail, we hope you find this sufficient.*

Discussion: Sections 5.1-5.3 are very short and do not add much discussion of the previous parts. I'd argue that they could simply be appended to their appropriate section in the results. Section 5.4 has some intriguing points, but is underdeveloped and feels a bit like an afterthought.
*Thank you for your thoughts on the discussion. As a result of reviewer comments, we have revisited the modelling (using WAIS divide as the reference accumulation rate record) and have shown the recently processed radar data from Icebridge flyovers on the island which we have added as further evidence for the conclusions we make in section 5.4. We have merged the sections 5.1 to 5.3 into one section (lines 287-314) with some small additions. We have added a new section which elaborates on estimates of accumulation rate and developed the "oldest ice" section further.*

Data availability: Are the data (*d*2H, chemical species, etc) that led to the creation of the age model in an online archive or otherwise available? I didn't see a link at Rowell 2022 (unless they are contained in that paper's supplementary material).

*You are right to point this out. We are also in the process of writing papers which will present and interpret the full, dated, stable water isotope and chemistry records and would like to publish them as standalone datasets when these papers are published, rather than there also be an additional dataset of just the top ~70 m of data, as we think this would be confusing and not necessary. We have uploaded the full dataset to Pangaea (reference PDI-35206) with an embargo until the end of 2023, to enable us to publish our initial findings from the dataset. We hope this seems like a reasonable solution to you and want to reiterate that we have no intention of not releasing the data for wider use.*

**Technical comments:**

17: The acronym MISI is never used again in the manuscript, and therefore it is not necessary to define it here.
*The acronym has been removed.*

21: Both of these acronyms are only used once elsewhere. Remove acronyms and just spell out in second instance.
*The acronyms have been removed.*

26: LIG is used three other times. It may be better for readability to simply spell it out each time.
*Due to expansion of the introduction and discussion, LIG is now used more frequently throughout the paper, so the acronym has been kept in the abstract and what is now line 32.*

45: Suggestion to consider flipping the sentences in this section so that you give a brief summary of the important necessary information here, and then end with the sentence that "a detailed description…".
*We have made other changes to this section as per your previous suggestion, so this is no longer needed.*

Section 3: The numbering scheme is odd with 3.0.1, 3.0.2, etc. Since there is no 3.1, shouldn't these simply be sections 3.1, 3.2, etc.?
*Yes thank you for pointing this out, the numbering has been corrected.*

67: Is SI core the name of the ice core, or an abbreviation for Sherman Island? If it is the core name, perhaps adding a date and/or length to the end will make it more unique of an identifier. If it is an abbreviation, Sherman Island isn't abbreviated elsewhere (which I think is good).
*Yes, SI:Core is the name we use to denote the 20 m firn core, to separate it from SI:RAID, which is named as such to make it clear we are talking about a "core" of RAID chippings, not a solid ice core. We name the RAID records from Little Dome C in this manner too (LDC:RAID1 and LDC:RAID2 for the two RAID boreholes which exist). For consistency and clarity over the type of ice we are referring to, we would like to maintain this naming system. You are right to point out that this is the first introduction of the naming convention in this*

*paper, however, so we have introduced "SI:RAID" earlier in the text (now line 88) and clarified the naming of the firn core at line 101 (previously 67).*

73: This phrasing seems odd to me. Perhaps "This variability could reflect the local geography of Sherman Island…"
*Agreed, upon reflection the intended meaning was not clear, we have taken your suggestion.*

74: "and IS then closely by…"
*Corrected.*

Figure 2 (and others): The line thicknesses are rather thick, which makes it hard to see small details where exactly indicated lines are falling. Consider making the thicknesses thinner.
*We have reduced the thickness of the lines in Figures 2, 3 and 4.*

102: Sentence starting with "That is" is a fragment.
*Corrected.*

111: a priori not prior, I think here.
*Due to another reviewer's comment, this sentence has been moved and re-worded so this phrase is no longer needed.*

Figure 4 (and others): This particular shade of red and green is difficult for many colorblind people. The colors aren't overlapping, so it isn't as big of a concern, but something to consider if you revise your figures. However, someone colorblind would not be able to make the connection from your legend to the lines.
*Thank you for pointing this out. We have labelled the lines in Figure 4 (Figures 2 and 3 have panel labels) so this should clarify which line is which.*

178: This paragraph could be merged with previous.
*Agreed, along with more detail being added to this section to clarify the difference between the models used, we have merged some of the paragraphs of this section.*

Fig 7: Is this the best x-axis display for this plot? It seems to add more confusion and oddness since the yr BP axis goes negative and positive on the log axis.
*Thank you for this comment. Presenting the age as relative to 1950 necessitates presenting negative values, and due to the desire to present this modelled data on a log x-axis scale to be able to visualise the range in the modelled outputs at an appropriate scale, this meant adding the negative values on the x-axis to -100. However, you are right that this could be confusing, so we have revisited the scale used here and have presented the data as relative to 2020 (meaning no negative values).*

**General Comments:**

There are a couple of areas within the manuscript where I felt lost and needed to read Rowell et al, 2022, for context. These areas are discussed in the Specific Comments below. If these areas are addressed, it will make the manuscript much more accessible to readers unfamiliar with Rowell et al, 2022, and help the manuscript achieve a "stand-alone" status.

With minor revisions, this paper will be an important contribution to the *Ice core science at the three poles* special issue.

*Thank you for your careful consideration of this manuscript and for your helpful suggestions. We agree that more information is required in the introduction and methods to ensure that the text can be considered a standalone paper without the need to refer to Rowell 2022. We have made additions to the introduction to provide more context, particularly with regard to the choice of Sherman Island as a drill site, and the use of the RAID, as you suggest in "specific comments". We have added some of the information that is in Rowell 2022 to the introduction and methods, specifically regarding how the ice was sampled and what makes RAID samples different from "normal" ice core samples (as also requested by another reviewer).*

**Specific Comments:**

35: Without the context from Rowell et al, 2022, it is not clear why WACSWAIN would choose to drill at Sherman Island to investigate the WAIS during the LIG if there is high risk of no LIG ice at the site. A sentence (or two) further explaining why Sherman Island was chosen as the site to investigate the LIG will help the reader understand the scientific context for drilling at Sherman Island.
*Thank you for pointing this out. We have expanded on the introduction and in what is now lines ~34-44, added more information about why Sherman Island was chosen. Specifically, we have explained that if LIG ice was present at Sherman Island it would provide an additional constraint into LIG WAIS stability to other cores from around the WAIS (Skytrain, soon to be Hercules Dome), providing an additional point for better spatial understanding of this time period.*

35: Without context from Rowell et al., 2022, it is not clear why the RAID was chosen for the drilling. All that is needed here is some of the context that is contained in Rowell et al., 2022, such as …high risk of not finding LIG ice at Sherman Island… the RAID was used, as opposed to a conventional ice core drill, to obtain a lower resolution water isotope record from the site which would indicate the ice age at the bedrock and give a first indication of the climate signal at Sherman Island… the RAID will save on logistics… etc..
*Yes this is useful information to include here as well as in the Rowell 22 paper, to save the reader having to look elsewhere. We have added more information about why the RAID was used on Sherman island. Specifically, that it was used because of the risk presented by Sherman island (of not containing LIG ice) so that in the event of not obtaining LIG ice, a full scale drill campaign would not have been carried out, but we would still have a long and useful record from a new and interesting location in the WAIS. We explain that the RAID is*

*much quicker than traditional drilling methods so this is essentially a compromise for a higher-risk site. Please see lines 44-56.*

40: The first sentence in the last paragraph talks about the "last few centuries" whereas the previous paragraph focuses on the LIG. I believe I understand your overall point – a reliable age scale is needed for the core – but the last paragraph could use some rewording to make it fit better within the context of all of the previous text up to that point.
*Thank you for your input on this. The aims of the WACSWAIN project and background for drilling at the site necessitate that we discuss at least briefly the LIG and why Sherman was chosen (which we have now expanded upon since the original submission). But you are right to also point out the abruptness of the change in focus from the LIG to the last few centuries. We have added some more text to link the overarching project aims with what we actually aim to achieve with this paper. Please see lines 57-64.*

60: It would be good to include a reference regarding "sulfur (S) isotope analysis to differentiate between background and volcanic samples".
*A reference to Patris et al 2000 has been added (line 82).*

**Technical Corrections:**

30: The word "in" is not needed in the sentence that reads, "the Abbott Ice Shelf in between continental Antarctica and Thurston Island".
*The word "in" has been removed*

45: The acronym for RAID is defined earlier in the text (at 30). Instead of "Sherman Island Rapid Access Isotope Drill (RAID) field campaign", it can simply say "Sherman Island RAID field campaign".
*This has been corrected.*

55: The title of this section is "SI:RAID age scale development". It will help the reader if "SI:RAID" is defined somewhere. The first sentence of this section could be changed from "The Sherman Island RAID age scale" to "The Sherman Island RAID (SI:RAID) age scale" to give a reader a clear understanding of what is meant by "SI:RAID".
*Thank you, yes this change has been added to (1) address your comment and (2) distinguish the SI:RAID from the SI:Core (the 20 m firn core) which is defined in the subsequent sentence in the text. The title of this section has been re-named to simply "Age scale development" to avoid confusion.*

65: Up to this point in the text, Sherman Island has not been assigned the acronym "SI". So, in the second sentence, it is clearer if "20 m long SI ice core (SI:Core)" is changed to "20 m long Sherman Island ice core (SI:Core)".
*Yes this is a helpful comment and your suggested change has been made.*

70: In some places throughout the text "sea salt" is used and in other places "sea-salt" is used.
*Occurrences of "sea-salt" have been changed to "sea salt".*

105: In the first sentence, include the word "Island" after "Sherman".
*Thank you, this is corrected.*

110: It seems like the first paragraph (both sentences) "It became clear during….dating of the deeper samples" should be in the previous section, and section 3.03 should begin with the sentence/paragraph "The records of chemical species were closely…".
*You are right that the sentence does make more sense at the end of the previous paragraph. We had placed it as the start of the "volcanic horizon identification" section to justify using the flank age model as the age scale to use for searching for volcanic horizons. This meaning still stands at the end of the previous section. The phrase "described below" has been added to the end of the sentence.*

110: Suggest rewording the second sentence to: "We use the flank-flow depth/age model to guide us in dating the deeper samples."
*Thank you, this is a clearer sentence and your suggestion has been taken.*

130: The authors are asked to double-check their stated values in the text for Rm and Rt. I think the values stated in the text are mismatched. I believe Rm should be 0.038 and not 1.78, and Rt should be 1.78 and not 0.038.
*Yes you are correct and this is an error. Rm and Rt have been swapped in the sentence to match with the correct values.*

170: Incorrect use of "an". Should read "we use a depth/age model".
*Corrected*

175: Should be either "Under these assumptions…" or "Under this assumption".
*Corrected to "these assumptions"*

200: Delete the first occurrence of the word "were" in the following sentence "Peaks present in some species in RAID data were which were not identified in the Core". Sentence should read, "Peaks present in some species in RAID data which were not identified in the Core"
*We have corrected the sentence as per your suggestion and added "the" - "Peaks present in some species in **the** RAID data…"*

210: It looks like the word "Table" is missing in "…added to the average uncertainty at the two adjacent ties (2)". Should it read "…added to the average uncertainty at the two adjacent ties (Table 2)".
*Corrected*

225: Suggest rewording sentence to "The model was used to estimate the age of the ice towards the bed at the RAID drilling site; however, in the lowest 13 meters, model outputs are meaningless."
*Along with a more detailed explanation of the different models, this sentence has been edited from the original.*

265: Previously in the text it is written as "S isotope" rather than "S-isotope".
*Corrected all occurrences to "S isotope"*

300: "IC" is defined anywhere in the text. Could use "chemistry data" to be consistent with Section 2 of the manuscript.
*Corrected to "chemistry data"*

300: Most previous places in the text it is written as "S isotope" rather than "S-isotope".
*Corrected all occurrences to "S isotope"*

305: "SI" isn't defined anywhere else in text. Suggest rewriting sentence to "The records contained in the existing Sherman Island data"
*Corrected to say "Sherman Island"*

310: "SI" isn't defined anywhere else in text. Suggest rewriting sentence to "The Sherman Island ice core was drilled by DT."
*Corrected both occurrences in the contributions to say "Sherman Island firn core" to make it very clear that this is referring to a separate core, not the RAID borehole and chippings.*

325: The Basen et al., 2012, reference is to a discussion paper. Is there a corresponding final paper to reference? If not, a reference to a published, non-discussion paper should be used in lieu of this one.
*Thank you for pointing this out, although this reference has now been removed as we use the WAIS Divide as a reference accumulation record (not EDC).*

340: The stated reference is to a pre-print. Printed manuscript reference is: Crick, L., Burke, A., Hutchison, W., Kohno, M., Moore, K. A., Savarino, J., Doyle, E. A., Mahony, S., Kipfstuhl, S., Rae, J. W. B., Steele, R. C. J., Sparks, R. S. J., and Wolff, E. W.: New insights into the ∼ 74 ka Toba eruption from sulfur isotopes of polar ice cores, Climate of the Past, 17, 2119–2137, https://doi.org/10.5194/cp-17-2119-2021, 2021.
*Reference updated*

**General comments**
This is a great paper, and the authors are to be commended! They have written the first paper to convincingly demonstrate the utility of the RAID system. They have also provided a really impressive and careful analysis of the timescale for the Sherman Island site, which will prove useful as a climate record, and strongly supports the idea of drilling a complete "normal" ice core record at this site.
*Thank you for your kind words and your assessment that both the RAID system is convincing and that the age scale for the Sherman Island RAID data is adequate. I am happy to hear that you agree Sherman Island would make a good future drill site!*

**Technical comments**
I have a few criticisms that I suggest should be considered in a revised paper.

First, a few minor grammar/style things:

Please define abbreviations before using them. In the asbstract, "LIG" is used, but not defined.
*LIG abbreviation now defined in abstract, and additional missing definitions elsewhere in the text have been identified and defined*

The terms SI:RAID and SI:CORE are used but not defined. At first, I thought SI might mean Supplementary Information!
*Thank you for pointing this out, both terms (SI:RAID and SI:Core) have now been defined in the text when they are introduced. I will steer clear of any use of "SI" on its own for this reason.*

In the abstract, it is stated that the new record is the "oldest, continuous, ice-derived palaeoclimate records for the coastal Amundsen-Bellingshausen Sea sectors." This is a stretch. WAIS Divide gets its snow accumulation from the Amundsen Bellinghausen Seas, and probably reflects the Amundsen-Bellingshausen Sea sectors almost as well as the Sherman Island site does. Instead, I would say "oldest, continuous, ice-derived palaeoclimate records *IN* the coastal Amundsen-Bellingshausen Sea sector.
*Thank you, you are right that is an important clarification. We have made the change you suggest.*

On Page 1, where Joughin and Alley 2011 are cited, I suggest a more primary source, rather than a review paper.
*Thank you, we have provided a reference instead to Joughin et al, 2014 https://doi.org/10.1126/science.1249055*

Similarly, on page 2, I don't think Deconto and Pollard is the best citation for the statement that the WAIS may have collapsed in the past. It would be better to cite evidence for this from sea level records or other papers that provide data suggestive of this. Deconto and Pollard is a modeling paper that *assumes* WAIS has collapsed. It is not a source of evidence that it did.
*Thank you, this is a good point. The Deconto and Pollard paper is useful for helping to set the context of why investigating the WAIS during the LIG further using empirical (ice core)*

*evidence is important, but it is not the correct citation for the statement here. I have instead referred to Dutton et al 2015. I have amended the subsequent sentence to "WACSWAIN… aims to use ice core records to investigate the WAIS during the LIG, to supplement existing modelling studies (e.g. Deconto and Pollard)" (lines 29-31).*

**Major comments**

On page 4, the model of Martín et al. (2015) is briefly described. But I would like to understand the motivation for using this model, rather than simply models such as that of Dansgaard-Johnsen, which is commonly used. Furthermore, elsewhere in the paper other models (e.g. Liboutry) are used, so this is confusing. Also, it is stated that horizontal advection is neglected, but nothing is said about whether this is a reasonable assumption for this site.
*Thank you for your helpful comment. As I explain at the top of this document, we have revisited the modelling work and, we think, improved the explanation of the modelling in the text (sections 2.2.2 and 2.2.4 and discussion).*

*In terms of horizontal advection, there is no empirical data for us to confirm or deny whether this is a reasonable assumption. Given this limitation and our consideration of all other input parameters, we choose to leave this assumption in place, however you are right to point it out. We think that for our needs of dating the deepest ice samples, and for giving a prediction (with quite large range) of the age of deeper ice available on the island, the assumptions are sufficient.*

On page 5, I find the Caption for the graph to be confusing. It is stated that divide flow is shown in red, flank flow in blue, and that green shows the optimized model. This implies that the best model is inconsistent with both divide flow and flank flow. But in the text, it's clear that this is NOT the difference between the curves. The difference is that assumptions made about the snow accumulation rate were probably too simple. That's a nice result, but it is not at all obvious. As far as I can tell, the green curve actually uses divide flow, but allows for variable accumulation. (If I am wrong, then I am confused by the text, and some work is needed to make it clear what has been done). I strongly suggest renaming the curves, with something like "divide flow with Dome C-proportional accumulation", "flank flow with Dome C-proportional" and "divide flow (??) and optimized accumulation".
*I assume this refers to Figure 5? We have changed the figure caption to, we hope, be more clear. You are correct that the green curve allows for variable accumulation, but in the updated manuscript is using a mean of the flank flow simulations (p = 1to4). We have renamed the lines to reflect which model they are showing, denoting them as the "forward model" and "optimised". We appreciate your detailed feedback on this figure.*

I have one significant criticism. Why is Dome C used for accumulation rate in the first place? It is about as far from Sherman Island as possible. Why not use WAIS Divide? Even South Pole would be better! The authors show clearly that the assumption that the accumulation history is proportional to that at Dome C does not work -- if it did, then the "optimized" timescale would be very similar to the "divide" timescales. There is a missed opportunity here. The authors do not show what accumulation history goes with the optimized timescale (they should show that!). It would be very interesting to know whether that history agrees better with Dome C or with WAIS Divide or South Pole.

I would very much like to see the "divide" and "flank" timescales re-calculated, using WAIS Divide (and perhaps also South Pole), and some commentary on which one provides the closest fit to the optimized timescale.

*Thank you for this assessment. I address these two comments together as they are concerning the same criticism. Another reviewer also raised concerns about the use of the Dome C accumulation rate as the reference. As we describe in the paper and at the top of this document, we have re-done the modelling work to address these concerns. As we show in the two following figures of modelled accumulation rate at Sherman Island, and at EDC and WD, changes in the optimised age/depth for SI:RAID for the time period over which tie markers exist is driven by the given time points at which accumulation rate varies and by the tie points. After the last tie marker, the accumulation rate is relative to WAIS Divide. The updated Figure 5 in the manuscript is similar in its conclusions to the previous paper, indicating that the site used for the accumulation rate reference record is only relevant beyond the last tie marker.*

***Accumulation rate over 25kyrs: EDC and WAIS Divide (top) and SI:RAID (bottom) showing different values of p (flank flow), and smoothed (smooth = yes) and unsmoothed (smooth = n/a). The unsmoothed data are used for the age scale, as they allow a good fit with the known tie markers. The smoothed data are used for the prediction of oldest ice available at summit of Sherman Island.***

[Figure]

*Accumulation rate over 1.5 kyrs: EDC and WAIS Divide (top) and SI:RAID (bottom) showing different values of p (flank flow), and smoothed (smooth = yes) and unsmoothed (smooth = n/a). The unsmoothed data are used for the age scale, as they allow a good fit with the known tie markers. The smoothed data are used for the prediction of oldest ice available at summit of Sherman Island.*

---

## Author Response (AR2)

**Editor**

Congratulations to the authors on the production of a very nice paper and thanks for the high quality, major revision they undertook along the way. Both reviewers 2 and 3 have suggested some final edits. The technical ones should be implemented - the optional ones I leave to the authors' discretion. I also note an error in the manuscript, which does not use the correct units for accumulation rate (depth/year or mass/area/year) - the authors have omitted the time unit. Please either change the unit, or ensure that the surrounding text refers to "annual accumulation of ". Also please ensure that depth units are clearly expressed in water- or ice-equivalent.

*Thank you for your comments. In the discussion section "Insights from the SI:RAID age scale", we have changed the text to refer to annual accumulation rates. This is the section where absolute rates of accumulation are discussed and compared.*

**Reviewer 2**

I find that the authors' substantial changes to the introduction and discussion have greatly improved the paper, particularly in making the paper have a stronger sense of purpose and impact. The RAID system and its pros/cons are clearly outlined now, and the discussion highlights the more unique aspects that RAID and this study bring to the broader Antarctic research field. I only have a few very minor possible technical changes for the authors to consider, but these are easily addressed in editing before final proofing.

*Thank you for your assessment and for reviewing the paper a second time. We are happy that you find the changes to be sufficient.*

36: Comma not needed after "stability"
*Fixed*

34-57: This is a long paragraph that could be split, probably at line 46 with the introduction of RAID.
*Paragraph split into two as you suggest*

46: RAID acronym should be defined here again in parentheses?
*RAID defined again at this point*

193: This sentence is very similar to 167. The context is slightly different, but it probably isn't necessary to list the volcanic events in the first instance if they are going to be repeated in the second instance here.
*Sentence in line 167 which lists the events has been removed.*

209: Is this a less than or equal sign? There's an actual symbol for this rather than a <=. Also at 329. Ignore if I am mistaken.
*Yes you are correct, now fixed in both locations.*

**Reviewer 3**

The authors have fully addressed my suggestions, and I support publication more-or-less as is.
*Thank you very much for reviewing the paper again and for your support.*

One minor suggestion which the authors may ignore if they wish!
In the following sentence:

Furthermore, a LIG record from this site, in addition to those from Skytrain Ice Rise and the upcoming Hercules Dome icecore further south (Fudge et al., 2022) and LIG data from the more westerly Mount Moulton (Korotkikh et al., 2011), would result in a more complete picture of the WAIS from this time.

It would be appropriate to cite Duestch et al. 2023 along with Fudge, and Steig et al. 2015 along with Korotkikh. Those two papers make the case that those specific sites (Duestch - SkyTrain, and Steig - Moulton) provide constraints on WAIS. Furthermore, the full Moulton record was not published in Korotkikh, but first appears in Steig. DOIs are 10.1175/JCLI-D-22-0647.1 and 10.1002/2015GL063861

*Citations added, thank you for pointing this out.*